# Transformation of valence signaling in a mouse striatopallidal circuit

**Donghyung Lee, Nathan Lau, Lillian Liu, Cory M Root\***

University of California San Diego, Department of Neurobiology, School of Biological Sciences, San Diego, United States

## eLife assessment

This **important** study by Lee and colleagues examined how neural representations are transformed between the olfactory tubercle (OT) and the ventral pallidum (VP) using single neuron calcium imaging in head-fixed mice trained in classical conditioning. They show that the dimensionality of neural responses is lower in the VP than in the OT and suggest that VP responses represent values in a more abstract form at the single neuron level while OT contains more odor information, potentially enhancing odor contrast. The results are overall **convincing** and this study provides insights into how odor information is transformed in the olfactory system.

**\*For correspondence:**
cmroot@ucsd.edu

**Competing interest:** The authors declare that no competing interests exist.

**Abstract** The ways in which sensory stimuli acquire motivational valence through association with other stimuli is one of the simplest forms of learning. Although we have identified many brain nuclei that play various roles in reward processing, a significant gap remains in understanding how valence encoding transforms through the layers of sensory processing. To address this gap, we carried out a comparative investigation of the mouse anteromedial olfactory tubercle (OT), and the ventral pallidum (VP) - 2 connected nuclei of the basal ganglia which have both been implicated in reward processing. First, using anterograde and retrograde tracing, we show that both D1 and D2 neurons of the anteromedial OT project primarily to the VP and minimally elsewhere. Using two-photon calcium imaging, we then investigated how the identity of the odor and reward contingency of the odor are differently encoded by neurons in either structure during a classical conditioning paradigm. We find that VP neurons robustly encode reward contingency, but not identity, in low-dimensional space. In contrast, the OT neurons primarily encode odor identity in high-dimensional space. Although D1 OT neurons showed larger responses to rewarded odors than other odors, consistent with prior findings, we interpret this as identity encoding with enhanced contrast. Finally, using a novel conditioning paradigm that decouples reward contingency and licking vigor, we show that both features are encoded by non-overlapping VP neurons. These results provide a novel framework for the striatopallidal circuit in which a high-dimensional encoding of stimulus identity is collapsed onto a low-dimensional encoding of motivational valence.

## Introduction

Animals exhibit an impressive ability to change how sensory inputs map onto behavioral outputs. Understanding how animals learn to output different behaviors through experience is one of the fundamental problems in neuroscience. Over the last half century, the field has developed compelling frameworks to tackle this problem at both the algorithmic level (Rescorla-Wagner models, Q-learning models; *Rescorla, 1972*; *Sutton, 1988*) and the mechanistic level (Hebbian learning, STDP, neuro-modulation; *Dan and Poo, 2004*). By comparison, we lack frameworks through which to understand how the brain might implement learning algorithms through the updating of synaptic weights. One

strategy has been to identify neural correlates of latent features assumed to be required for these algorithms (e.g. dopamine as a neural substrate for reward-prediction-error; *Hollerman and Schultz, 1998*; *Schultz et al., 1997*). These results, however, can often be difficult to interpret because reward related signals are found globally throughout the brain (*Allen et al., 2019*), and are likely multiplexed with signals about motor output and/or stimulus identity. We propose that a more powerful approach is one that compares (1) how the encoding of reward cues changes from one brain nucleus to its downstream target and (2) how much of the encoding can be explained by valence vs. other features such as identity or motor output. In this present work, we implement this comparative approach to the investigation of how encoding of olfactory reward cues is transformed between the olfactory tubercle (OT) and the ventral pallidum (VP) in the context of classical conditioning.

The OT, also known as the tubular striatum (*Wesson, 2020*), is a three-layered striatal nucleus situated at the bottom of the forebrain. As with other striatal structures, the OT is composed primarily of Spiny Projection Neurons (SPN's) which express either the *Drd1* or *Drd2* DA receptors (abbreviated as $OT_{D1}$ and $OT_{D2}$, respectively; *Tritsch and Sabatini, 2012*). In addition to receiving a wide range of inputs from cortical and amygdalar areas (e.g. AI, OFC, BLA, PlCoA, Pir) (*Zhang et al., 2017b*), it receives dense DAergic input from the midbrain (*Ikemoto, 2007*) and direct input from the mitral and tufted cells of the olfactory bulb (*Haberly and Price, 1977*; *Igarashi et al., 2012*; *Scott, 1981*), a unimodal and primary sensory area. There is a range of experiments that suggest that the OT's DAergic innervation is involved in reward processing. Coincident stimulation of the lateral olfactory tract and DAergic midbrain afferents supports LTP of excitatory current (*Wieland et al., 2015*) and rats self-administer cocaine, a DAergic drug, into the medial OT more vigorously than to any other striatal nuclei (*Ikemoto, 2003*). And while the OT neurons are known to respond to a wide range of odorants (*Wesson and Wilson, 2010*), pairing stimulation of midbrain DAergic neurons with an odor drives appetitive behavior towards the paired odor (*Zhang et al., 2017a*) and enhances the contrast of the odor-evoked activity (*Oettl et al., 2020*). Lastly, a number of recent publications report varying degrees of valence signals recorded from neurons in the OT (*Gadziola et al., 2020*; *Gadziola et al., 2015*; *Martiros et al., 2022*; *Millman and Murthy, 2020*; *Oettl et al., 2020*).

The most well-established target of the OT is the VP (*Newman and Winans, 1980*; *Zahm and Heimer, 1987*), a pallidal structure that lies immediately dorsal to the OT. In addition to OT input, VP receives strong input from the nucleus accumbens (*Jones and Mogenson, 1980*) and the subthalamic nucleus (*Ricardo, 1980*; *Turner et al., 2001*). More recently, it was reported that VP also receives inputs from several cortical and amygdalar areas that also project to the OT (e.g. Pir, BLA, OFC; *Stephenson-Jones et al., 2020*). The VP contains GABAergic neurons, which respond to positive valence cues, and glutamatergic neurons, which respond to negative valence cues. Consistent with their responsiveness, the GABAergic and glutamatergic neurons drive real time place preference and avoidance, respectively (*Faget et al., 2018*). Although it is well-established that the VP plays a critical role in reward processing, there has been ongoing disagreement on what specific latent features are encoded by VP neurons. Interpretations have included valence (*Ottenheimer et al., 2018*; *Ottenheimer et al., 2020b*; *Richard et al., 2016*; *Tachibana and Hikosaka, 2012*), hedonics (*Smith et al., 2009*; *Tindell et al., 2006*), motivation (*Faget et al., 2018*; *Fujimoto et al., 2019*; *Lederman et al., 2021*; *Tindell et al., 2005*), and reward-prediction error (*Ottenheimer et al., 2020a*). This ongoing discussion highlights the need to adopt a more comparative approach outlined above.

Here, we investigated the transformation of learned association encoding between the OT and the VP. We began by refining our understanding of OT's efferents to reveal that, contrary to a previous report, both $OT_{D1}$ and $OT_{D2}$ neurons of the anteromedial OT project primarily to the ventrolateral portion of the VP and minimally elsewhere. Given this finding that VP may be the only robust output of the anteromedial OT, we proposed that the OT to VP circuit is an ideal model system for examining how the encoding of reward cues is transformed between connected brain areas. Comparing the stimulus-evoked activity in $OT_{D1}$, $OT_{D2}$, and VP neurons with two-photon $Ca^{2+}$ imaging, we found that VP neurons encode reward-contingency in low-dimensional space with good generalizability. In contrast, activity in both $OT_{D1}$ and $OT_{D2}$ neurons was high-dimensional and primarily contained information about odor identity, although $OT_{D1}$ neurons are modulated by reward. By examining the same neurons across multiple days of pairing, we propose a putative cellular mechanism for reward-cue responsiveness in VP wherein reward responsive VP neurons gradually become reward-cue responsive. Finally, using a novel classical conditioning paradigm, we provide evidence that non-overlapping

sets of VP neurons contain information about the vigor of licking and reward-contingency, but not both.

## Results

In order to compare odor-evoked activity in connected brain nuclei, we first characterized which specific subregions of the VP receive input from the anteromedial portion of the OT. While considerable effort has been made to unravel the anatomy and function of the NAc, much less attention has been directed at the OT. Although multiple studies have characterized its anatomical connectivity (*In 't Zandt et al., 2019, Zahm and Heimer, 1987*; *Zhang et al., 2017b*; *Zhou et al., 2003*), there is inconsistency regarding whether or not OT projects to areas other than the VP. We therefore aimed to clarify previously reported OT connectivity by independently conducting anterograde viral tracing experiments in $OT_{D1}$ and $OT_{D2}$ neurons of the anteromedial OT. To this end, we injected AAVDJ-hSyn-FLEX-mRuby-T2A-syn-eGFP in the anterior OT of *Drd1*-Cre (labels D1 +SPN's) and *Adora2a*-Cre (labels D2 +SPN's) animals (*Figure 1A*, *Figure 1—figure supplement 1C–E*). Because viral contamination of areas dorsal to the target site can lead to difficulties in interpretation of tracing data, we also injected the same virus to the AcbSh immediately dorsal to the OT for comparison. Consistent with past findings (*Kupchik et al., 2015*), we observed robust projections VP, LH, and VTA from $AcbSh_{D1}$ neurons and primarily VP projections $AcbSh_{D2}$ neurons (*Figure 1B–C*). We also observed dense labeling of the VP in D1-Cre and A2A-Cre animals injected at the OT. Contrary to one report (*Zhang et al., 2017b*) but consistent with another (*Zhou et al., 2003*), we observed minimal labeling in LH and VTA, or anywhere else in the brain, for both $OT_{D1}$ and $OT_{D2}$ experiments (*Figure 1B–C*, *Figure 1—figure supplement 1A, B*), suggesting that neither OT subpopulation from the anteromedial OT projects strongly outside the VP. As previously reported (*Groenewegen and Russchen, 1984*). It is also notable that OT projections were restricted to the lateral portions of the VP.

To corroborate and more precisely describe the OT to VP projection, we conducted retrograde tracing by injecting CTB::488 and CTB::543 to the lateral and medial portion of caudal VP, respectively (*Figure 1D*, *Figure 1—figure supplement 1F–G*). We found strong labeling of soma by both CTB::488 and CTB::543 in the Acb, AI, and Pir (*Figure 1E–F*). By comparison, we found predominantly CTB:488, but not CTB::543, labeling in OT soma, indicating OT neurons are more likely to project to the lateral portion of the VP than to the medial. Similarly, to corroborate the lack of OT to VTA projection, we injected CTB::647 into the VTA (*Figure 1G*, *Figure 1—figure supplement 1H*). Consistent with previous findings (*Beier et al., 2015*; *Faget et al., 2016*; *Watabe-Uchida et al., 2012*), we found dense labeling of soma in various areas of the striatum such as AcbSh, AcbC, and CPu (*Figure 1H–I*). We also found some labeling of soma in some frontal cortical regions such as PrL, AI, and IL cortices. In contrast, we found that hardly any neurons within any part of the OT were labeled. The rare OT neurons that did have CTB labeling were exclusively localized to the dorsal most portion of layer III, closely bordering the VP. Taken together, we conclude that both D1 and D2 SPN's of the anteromedial OT project primarily to the lateral portion of the VP and negligibly to other brain areas, including the VTA.

Once we had identified that the anteromedial OT has extremely constrained outputs to the lateral VP, we set out to comparatively characterize the encoding of reward cue in this striatopallidal circuit. Past analysis of valence encoding is confounded by not accounting for the difficult-to-avoid overlaps among identity, salience, and reward contingency. To address this, we carefully designed a 6-odor conditioning paradigm where these factors could be decoupled (*Figure 2A*). During each trial, the animal is exposed to 1 of 6 odors for 2 s. At the end of odor delivery, the animal either receives: 2 μl of a 10% sucrose solution (S), 50ms of airpuff at 70 psi (P) or nothing (X). 3 of the odors are ketones (hexanone, heptanone, octanone) and the rest are terpenes (terpinene, pinene, limonene), but the pairing contingencies are chosen such that each contingency group (S, P, or X) includes 1 ketone and 1 terpene. We reason that in a valence-encoding population, but not in an identity-encoding population, we should see that odor pairs of different reward-contingency (e.g. $S_K$, a sucrose-paired ketone vs. $P_K$, an airpuff-paired ketone) are more different than odor pairs of same reward-contingency (e.g. $S_K$, a sucrose-paired ketone vs. $S_T$, a sucrose-paired terpene). Additionally, because both sucrose-pairing and airpuff-pairing should make the associated odor more salient, we can disambiguate between increased discriminability due to salience vs. valence by comparing neural activity in response to sucrose-cues or airpuff-cues.

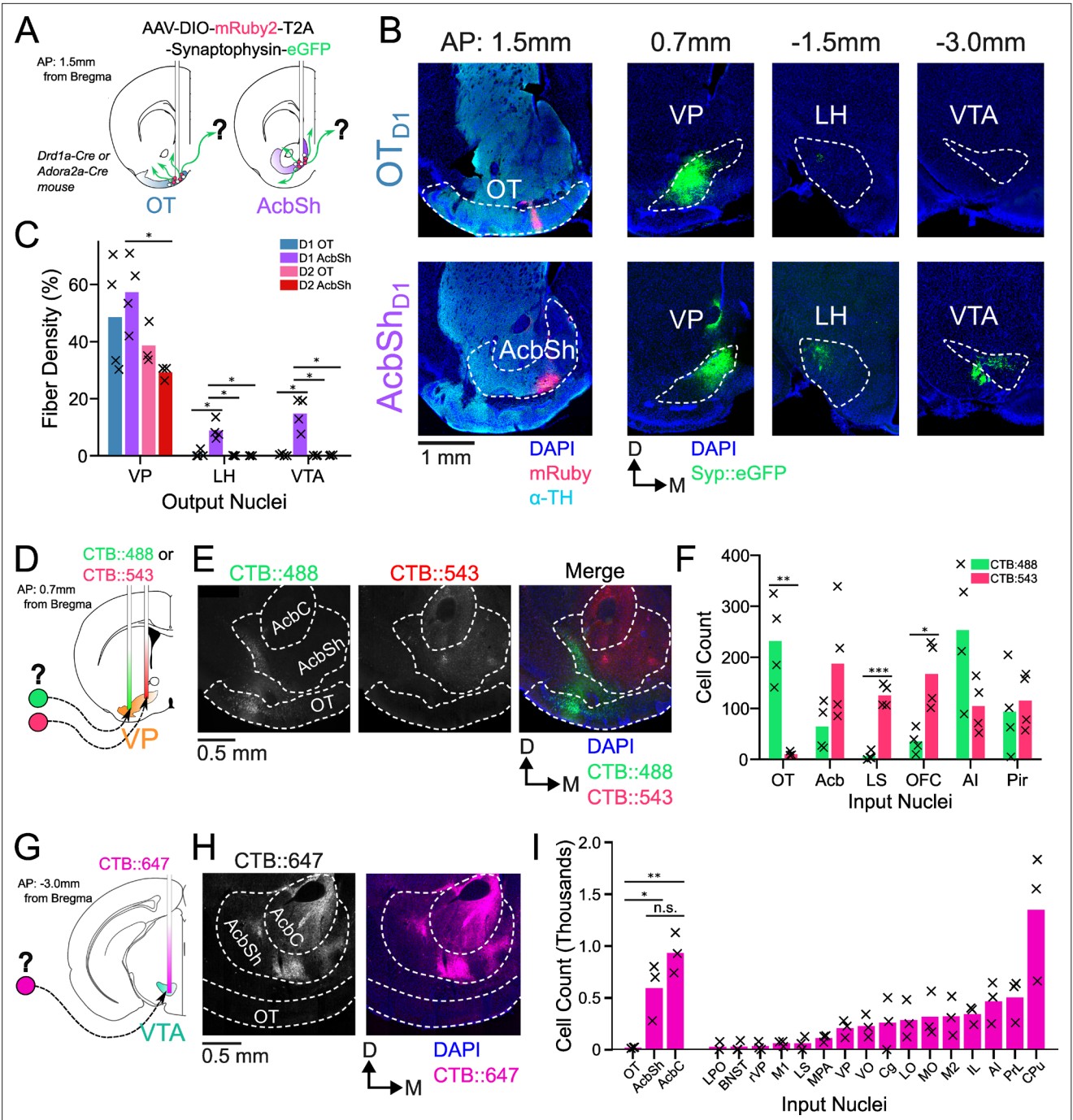

**Figure 1.** OT$_{D1}$ and OT$_{D2}$ primarily project to the lateral portion of the VP. (**A**) Schematic representation of Cre-dependent anterograde axonal AAV tracing experiments used to characterize outputs of OT neurons. *Drd1+* and *Drd2+* neurons were separately labeled by using *Drd1*-Cre and *Adora2a*-Cre mouse lines, respectively. (**B**) Representative images from OT$_{D1}$ (top) vs. the AcbSh$_{D1}$ injection (bot). Target sites (far-left column) are stained with α-tyrosine hydroxylase antibodies to visualize the boundary between VP and OT. (**C**) Quantifying the % of output regions with fluorescence (n=3–4). (**D**) Schematic representation of two-color retrograde CTB tracing experiment used to confirm OT to VP connectivity. CTB::488 and CTB::543 were injected to the lateral and medial portion of the VP, respectively. (**E**) Representative images of CTB labeled neurons in the OT and Acb. (**F**) The number of labeled cells was quantified (n=4). (**G**) Schematic representation of retrograde CTB tracing experiment used to test OT to VTA connectivity. CTB::647 was injected in the VTA. (**H**) Representative image shows robust AcbSh and AcbC labeling but no OT labeling. (**I**) Quantification of labeling in different nuclei (n=3). Pairwise comparisons were done using the Student's t-test. The p-values were corrected for FDR by Benjamini-Hochberg procedure. ***p<0.001, **p<0.01, *p<0.05. See *Appendix 1—tables 1–3* for detailed statistics.

The online version of this article includes the following figure supplement(s) for figure 1:

**Figure supplement 1.** OT$_{D1}$ and OT$_{D2}$ primarily project to the lateral portion of the VP.

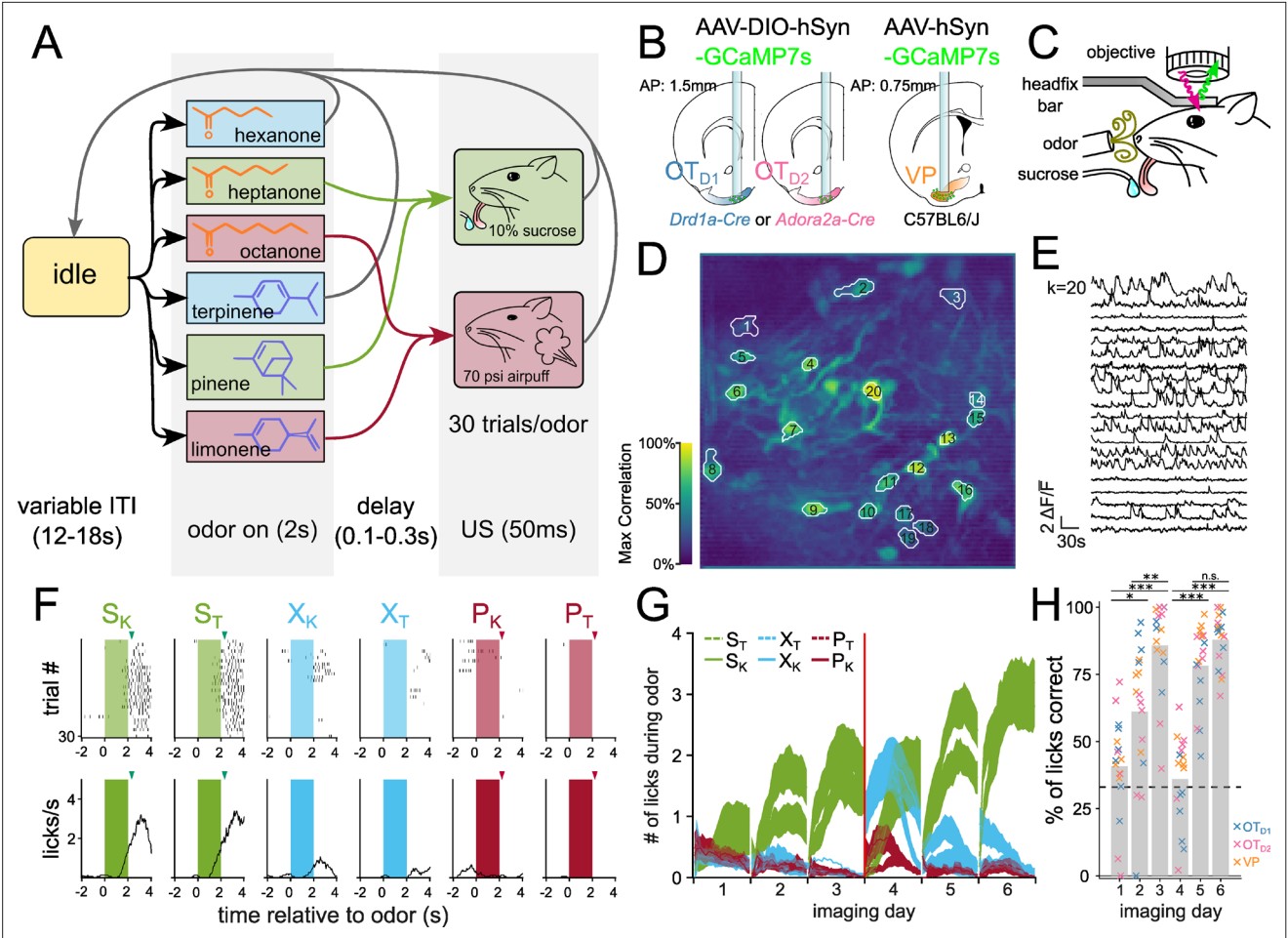

**Figure 2.** Head-fixed two-photon $Ca^{2+}$ imaging of $OT_{D1}$, $OT_{D2}$, or VP neurons during 6-odor conditioning paradigm. (**A**) State-diagram of odor conditioning paradigm. Each trial begins with 2 s of odor delivery. Odors are chosen in pseudorandomized order such that the same odor is not repeated more than twice in a row. At the end of odor delivery, there is a variable delay (100–300ms), after which the animal is given either a 10% sucrose solution ($S_K$ and $S_T$), a 70 psi airpuff ($P_K$ and $P_T$), or nothing ($X_K$ and $X_T$). Trials are separated by a variable intertrial interval (ITI; 12–18 s). Schematic representation of (**B**) lens implant surgery and (**C**) headfix two-photon microscopy setup. An example of spatial (**D**) and temporal (**E**) components extracted by CNMF from *Drd1*-Cre animal on day 3 of imaging. (**D**) The spatial footprints of 20 example neurons are shown on top of a maximum-correlation pixel image that was used to seed the factorization. The number displayed over each neuron matches the row number of the temporal components in (**E**). (**F**) An example raster plot (top) and averaged-across-trials trace (bottom) of the licking behavior recorded concurrently as (**D**) and (**E**). The timing of odor delivery is shown as shaded rectangles. The timing of US delivery is shown as arrowheads. (**G**) The mean total licks during each of the odors is shown averaged across all animals (n=17) after application of a moving-average filter with a window size of 10 trials. Red line marks the sucrose and airpuff contingency switch between day 3 and day 4. (**H**) Bar graph showing the licks during either sucrose cue expressed as a fraction of all licks during any odor. FWER-adjusted statistical significance for post hoc comparisons are shown as: ***p<0.001, **p<0.01, *p<0.05. See *Appendix 1—tables 4 and 5* for detailed statistics.

The online version of this article includes the following figure supplement(s) for figure 2:

**Figure supplement 1.** Histological verification of lens implant location.

**Figure supplement 2.** Pooled averaged-over-trials neural activity of all neurons from $OT_{D2}$ animals across days.

**Figure supplement 3.** Pooled averaged-over-trials neural activity of all neurons from $OT_{D1}$ animals across days.

**Figure supplement 4.** Pooled averaged-over-trials neural activity of all neurons from VP animals across days.

**Figure supplement 5.** Extended behavioral analysis from imaging period.

**Figure supplement 6.** Traces of example neurons and their corresponding metrics.

**Figure supplement 7.** Percentage of neurons responsive to each odor across days.

**Figure supplement 8.** Distribution of response magnitudes to each odor across days.

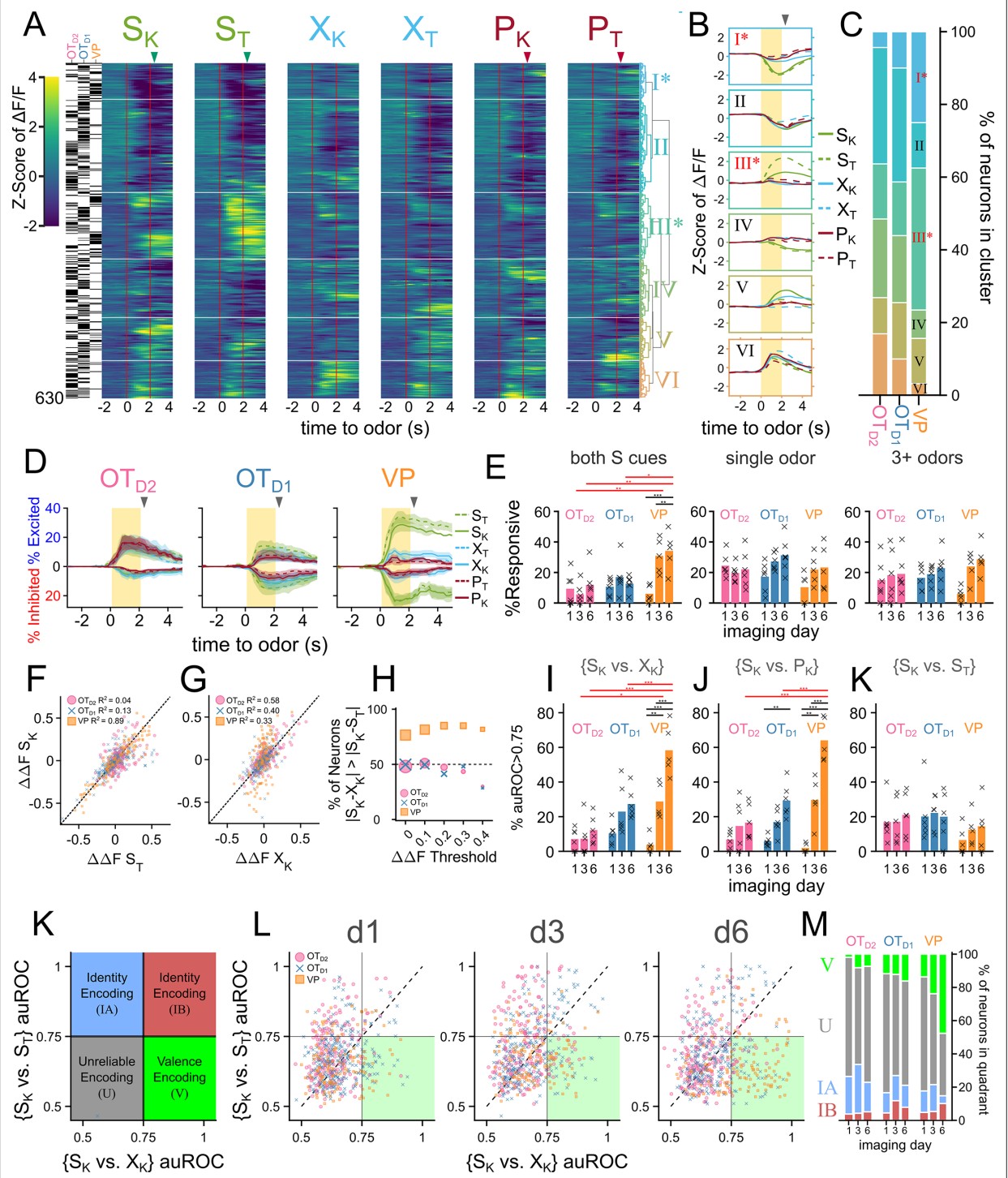

**Figure 3.** VP neurons encode reward-contingency more robustly than OT$_{D1}$ or OT$_{D2}$ neurons. (**A**) Heatmap of odor-evoked activities in OT$_{D1}$, OT$_{D2}$, and VP neurons from day 6 of imaging. The fluorescence measurements from each neuron were averaged over trials, Z-scored, then pooled for hierarchical clustering. Neurons are grouped by similarity, with the dendrogram shown on the right and a raster plot on the left indicating which region a given neuron is from. Horizontal white lines demarcate the boundaries between the 6 clusters. Odor delivered at 0–2 s marked by vertical red lines and US delivery is marked by arrowheads. From left to right, the columns represent neural responses to sucrose-paired ketone and terpene, control ketone and terpene, and airpuff-paired ketone and terpene (S$_K$, S$_T$, X$_K$, X$_T$, P$_K$, P$_T$). (**B**) Average Z-scored activity of each cluster to each of the six odors on day 6 of imaging. Yellow bar indicates 2 s of odor exposure. (**C**) The distribution of clusters by population. (**D**) Percentage of total neurons that were significantly excited or inhibited by each odor (Bonferroni-adjusted FDR <0.05) as a function of time relative to odor. Lines represent the mean across biological replicates and the shaded area reflects the mean ± SEM. (**E**) Bar graph showing % of neurons from each population that are responsive to

*Figure 3 continued on next page*

*Figure 3 continued*

both sucrose-paired odors in the same direction (left), responsive to only a single odor (middle), or responsive to at least 3 odors (right). Bars represent the mean across biological replicates and x's mark individual animals. (**F**) Scatterplot comparing the magnitudes of $S_K$ responses ($\Delta\Delta S_K$) to $S_T$ responses ($\Delta\Delta S_T$). The dotted line represents the hypothetical scenario where $\Delta\Delta S_K = \Delta\Delta S_T$. For each population, the $R^2$ value of the 2-d distribution compared to the $\Delta\Delta S_K = \Delta\Delta S_T$ line is reported. (**G**) Same as F but comparing $\Delta\Delta S_K$ to $\Delta\Delta X_K$. (**H**) Lineplot showing the % of neurons from each population where the difference between $\Delta\Delta S_K$ and $\Delta\Delta X_K$ is lower than that between $\Delta\Delta S_K$ and $\Delta\Delta S_T$. (**I**) Bargraph showing % of neurons whose responses to {$S_K$ vs. $X_K$} can be discriminated by a linear classifier with auROC >0.75. (**J**) Same as (**I**) but for {$S_K$ vs $P_K$}. (**K**) Same as (**I**) but for {$S_K$ vs $S_T$}. (**L**) Schematic representation of four possible categories for a joint-distribution of {$S_K$ vs. $X_K$} and {$S_K$ vs. $S_T$} auROC values. Identity-encoding neurons could be in any quadrant other than the bottom-left, whereas valence-encoding neurons should be in the bottom-right quadrant. (**M**) Scatterplot of each neuron's auROC value for {$S_K$ vs. $X_K$} on the x-axis and {$S_K$ vs. $S_T$} on the y-axis on days 1, 3, and 6 of imaging. (**N**) Stacked bar graph showing the distribution of neurons from each population that fall into each of the four quadrants across the 3 different imaging days. FWER-adjusted statistical significance for post hoc comparisons are shown as: \*\*\*p<0.001, \*\*p<0.01, \*p<0.05, n.s. p>0.05. See *Appendix 1—tables 6–17* for detailed statistics. Source data available at 10.5061/dryad.2547d7x15.

The online version of this article includes the following figure supplement(s) for figure 3:

**Figure supplement 1.** Pairwise analysis of single neuron odor encoding.

**Figure supplement 2.** Multinomial analysis of single neuron odor encoding.

To record the activity of the OT and VP neurons across multiple days of pairing, we injected C57BL/6 mice with AAV9-hSyn-jGCaMP7s-WPRE (lateral VP) and *Drd1*-Cre or *Adora2a*-Cre animals with AAV9-hSyn-FLEX-jGCaMP7s-WPRE (anteromedial OT; *Figure 2B*, *Figure 2—figure supplement 1A-F*). Additionally, we implanted a 600 µm Gradient Refractive Index (GRIN) lens 150 µm dorsal to the virus injection site and cemented a head-fixation plate to the skull. Six to eight weeks after surgery, animals were water-restricted and habituated for 3–5 days in the head-fixation setup (*Figure 2C*). We processed the acquired time-series images using Constrained Nonnegative Matrix Factorization (*Pnevmatikakis et al., 2016*) to obtain fluorescence traces from each putative neuron (*Figure 2D and E*). In total, we recorded Ca$^{2+}$ signals from 231 OT$_{D2}$ neurons from 6 *Adora2a*-Cre animals (*Figure 2*, *Figure 2—figure supplement 2*), 288 OT$_{D1}$ neurons from 6 *Drd1*-Cre animals (*Figure 2*, *Figure 3*) and 130 VP neurons from 5 C57BL6/J animals (*Figures 2 and 3*, *Figure 4*).

After 3 days of odor-sucrose associations, the animals displayed anticipatory licking behavior primarily during sucrose-paired odors (*Figure 2F and G*). Starting on day 4, the sucrose and airpuff contingencies were switched such that every odor had a reassigned contingency. By day 6, animals had adapted their anticipatory licking behavior to match the new sucrose-contingency (*Figure 2G*). Quantification of the animal's licking behavior showed that the accuracy of animals' licks during odor increased across time and was not different across lens-placement groups (*Figure 2H*, *Appendix 1—tables 4 and 5*; ANOVA: $F_{day}$ = 27.64, $p_{day}$ = 2.29e-16, $F_{lens\ location}$=2.30, $p_{lens\ location}$=0.11). These results show that the animals learn to associate S odors with reward in a flexible manner in our paradigm. Because we saw the strongest behavioral evidence that animals learned odor-sucrose associations by day 6, we focused our analysis on how reward cues are encoded on the last day of imaging. The animals also showed trends of behavioral changes in response to airpuff-cues, though they were not significant: during airpuff-cues, animals walked less and closed their eyes more than during other odors (*Figures 2–4*, *Figure 5D-G*, *Appendix 1—tables 32–36*). These behavioral changes for aversive cues were less robust than that for reward association. However, animals show clear responses to the US indicating that they perceive the aversive stimulus.

OT and VP neurons showed heterogeneous responses to 6 odors across all 6 days of imaging (*Figure 2—figure supplement 2*, *Figure 2—figure supplements 3 and 4*). To unbiasedly describe the difference between regions, we performed hierarchical clustering on the pooled trial-averaged responses to the 6 odors on the 6th day of imaging (*Figure 3A*). We observed both inhibitory (clusters I, II) and excitatory (clusters III-VI) responses to odors as well as broad (clusters II, VI) and narrow (clusters IV, V) odor-tuning (*Figure 3B*). Cluster I and cluster III most closely fit our description of putative valence-encoding neurons, that is neurons that had similar responses to 2 sucrose-cues ($S_K$ vs. $S_T$) but different responses to a sucrose-cue and a puff-cue or control odor ($S_K$ vs. $P_K$ or $X_K$). Although all clusters included neurons from all subpopulations, cluster I and cluster III, which showed larger responses to odors predicting sucrose, were enriched for VP neurons (*Figure 3C*), leading us to hypothesize that individual neurons in the VP were more likely to be valence encoding neurons than in either anteromedial OT subpopulation.

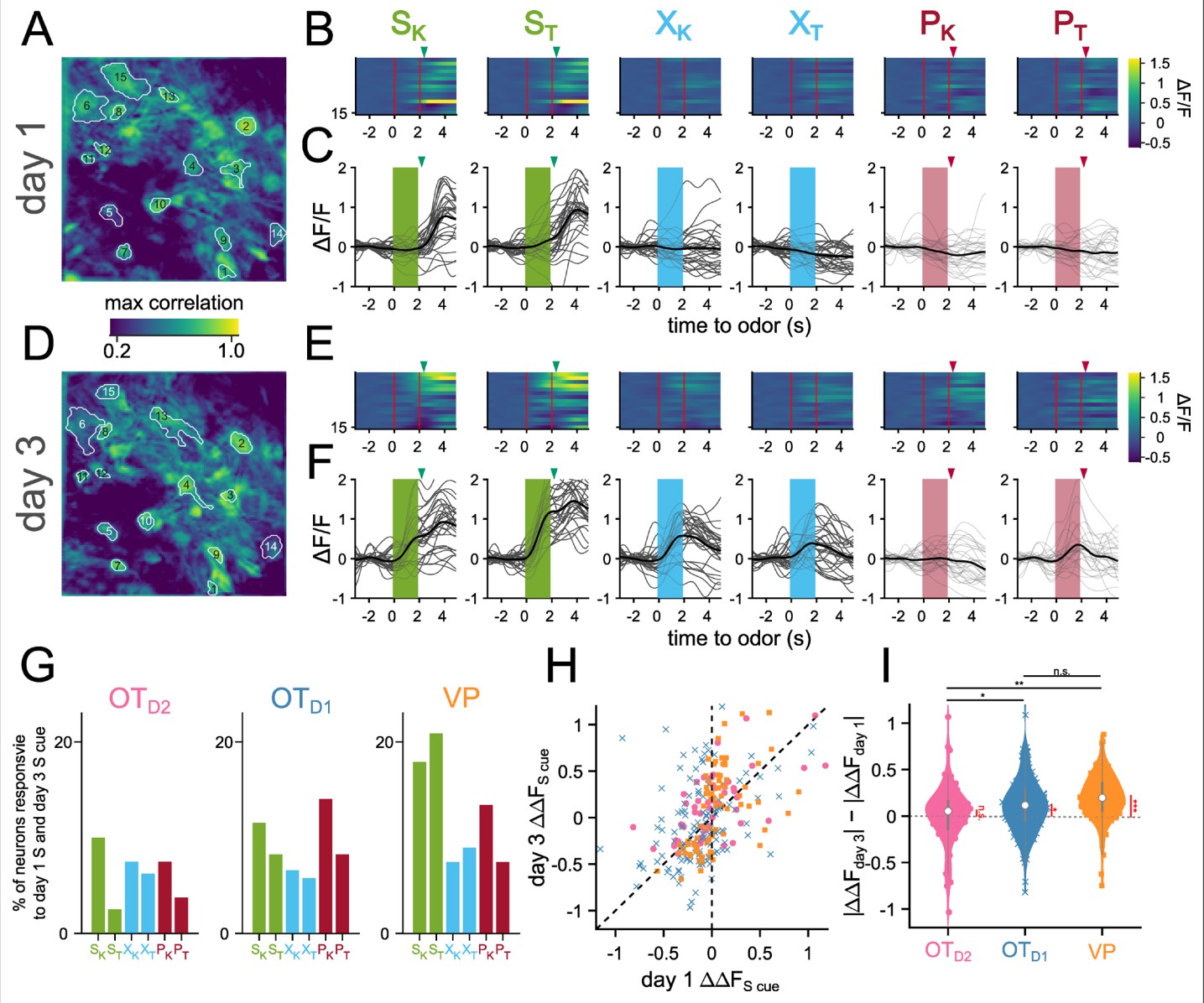

**Figure 4.** Sucrose responsive VP neurons become sucrose-cue responsive after pairing. (**A**) The spatial footprints of 15 neurons from day 1 are outlined over a max-correlation projection image. (**B**) Heatmap of averaged-over-trials ΔF/F in response to 6 odors on day 1. Odor delivery period is shown with 2 red vertical lines and sucrose/airpuff timing is shown with downward arrowhead. (**C**) An example neuron's responses on day 1 across 30 trials to 6 different odors. Individual trial traces are shown in light gray whereas the averaged-across trials trace is shown in black. Odor delivery period is depicted as shaded rectangles and US delivery is marked by arrowheads. (**D–F**) Same as (**A–C**), respectively, but for day 3. (**G**) Percentage of all tracked neurons that were both sucrose-responsive on day 1 and odor-responsive in the same direction on day 3. (**H**) Scatter plot of averaged-over-trials responses to $S_K$ or $S_T$ on day 1 (x-axis) and day 3 (y-axis). Each point is a neuron that was successfully matched from day 1 and day 3. Neurons from $OT_{D2}$, $OT_{D1}$, and VP are plotted as pink circles, blue crosses, and yellow squares, respectively. Neurons that have increased response magnitudes on day 3 would fall between the two dotted lines. (**I**) Violin plot showing the distributions of day 3 responsive magnitude – day 1 response magnitude. Black asterisks show statistical significance of pairwise comparisons and red asterisks show statistical significance for one-sample t-tests. Pairwise comparisons were done using the Student's t-test. The p-values were corrected for FDR by Benjamini-Hochberg procedure. \*\*\*p<0.001, \*\*p<0.01, \*p<0.05, n.s. p>0.05. See *Appendix 1—tables 18 and 19* for detailed statistics. Source data available at 10.5061/dryad.2547d7x15.

To assess this hypothesis, we quantified the number of neurons that had statistically significant responses to each of the 6 odors on the last day of imaging. We found that more VP neurons were either excited (29.8 ± 4.1%, 36.6 ± 4.0% for $S_K$, $S_T$) or inhibited (24.5 ± 3.0%, 29.4 ± 3.8% for $S_K$, $S_T$) to either sucrose-paired odor than to control or puff-paired odors (7.6–11.1% excited, 8.1–12.9% inhibited; *Figure 3D*, *Figure 2—figure supplement 7A*). For statistical comparisons see (*Figure 2—figure*

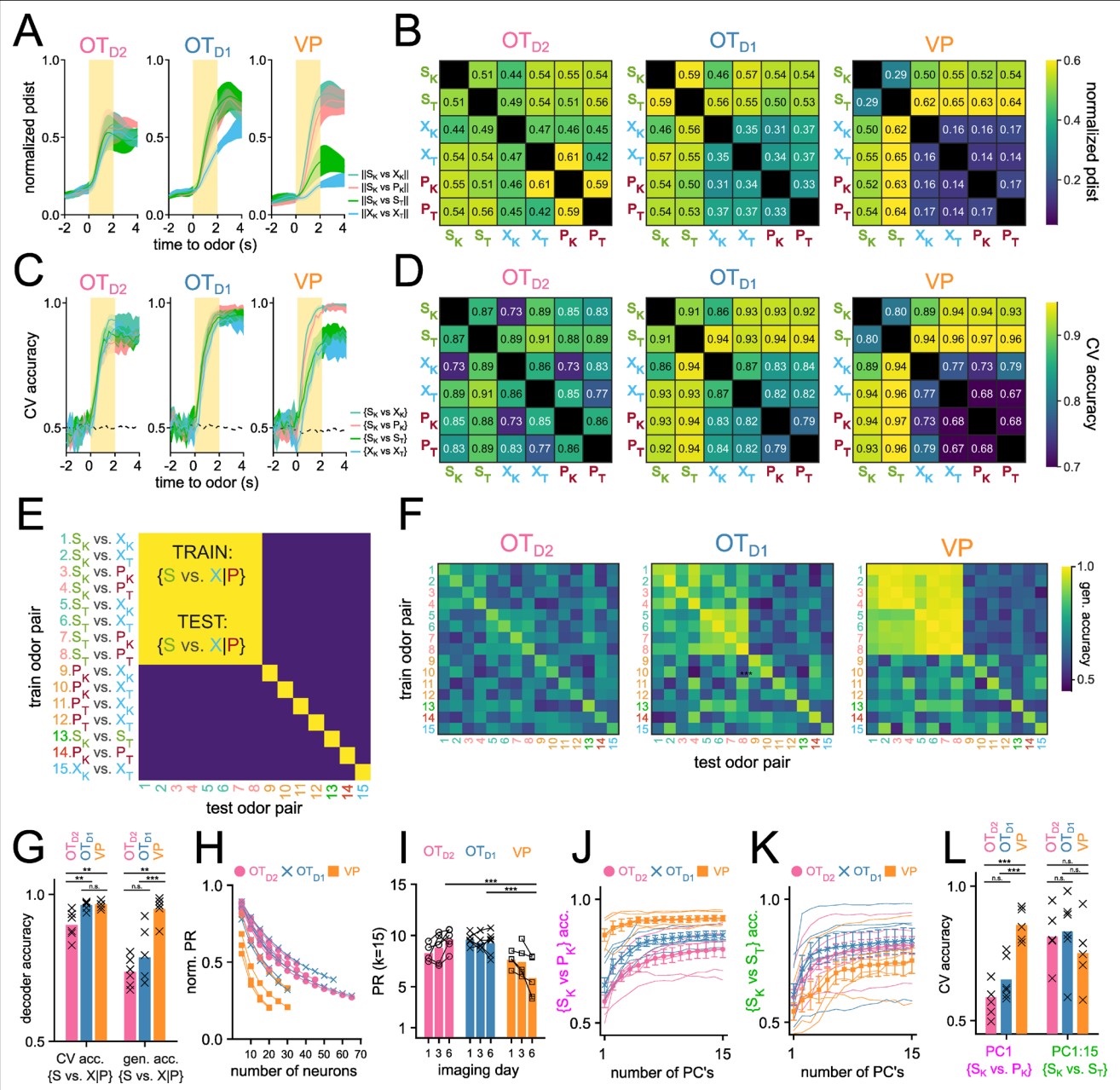

**Figure 5.** OT encodes odor identity in high-dimensional space and VP encodes reward-contingency in low-dimensional space. (**A**) Average normalized pairwise Euclidean distance between odor-evoked population-level activity from day 6 of imaging shown as a function of time relative to odor delivery. Traces show the average value across biological replicates of the same population and the shaded areas represent the average ± SEM. (**B**) A heatmap of the average normalized pairwise distance during the odor delivery period. (**C**) Average CV accuracy of binary pairwise linear classifiers trained on population data plotted against time relative to odor delivery. (**D**) A heatmap of the average CV accuracy during the odor delivery period. (**E**) Schematic representation of generalized linear classification performance for an idealized valence encoder. Each row corresponds to the training odor-pair and each column corresponds to the testing odor-pair. For an idealized valence encoder, the decodability would generalize well across odor-pairs of the equal valence grouping outlined in red. Note that the elements along the diagonal are cases where training and testing odor-pairs are identical and do not reflect generalizability. (**F**) Heatmap representing the maximum generalized linear classification accuracy during odor delivery period averaged across biological replicates for each population. (**G**) Mean cross-validated linear classifier accuracy for S-cue vs. control or puff-cue classification and the generalized accuracy for S-cue vs. control or puff-cue classification after training on a different pair. Bar represents the mean across biological replicates and x's mark accuracy values for individual animals. (**H**) Average PR normalized to *n* calculated after randomly subsampling an increasing number of neurons. (**I**) Average PR calculated after subsampling 15 neurons. (**J**) Average CV accuracy of linear classifiers trained on {$S_K$ vs. $P_K$} plotted against number of principal components used for training. For each simultaneously imaged group of neurons, 15 neurons were subsampled and classifiers were trained on an increasing number of principal components. Thinner faded lines show mean accuracy across subsampling for individual animals. Markers

*Figure 5 continued on next page*

*Figure 5 continued*

represent the mean across biological replicates. Error bars indicate SEM across biological replicates. (**K**) Average CV accuracy of linear classifiers trained on {$S_K$ vs. $S_T$}. (**L**) Comparison of the average accuracy of {$S_K$ vs. $P_K$} classifiers trained on the 1st PC vs. {$S_K$ vs. $S_T$} classifiers trained on all 15 PCs. FWER-adjusted statistical significance for post hoc comparisons are shown as: ***p<0.001, **p<0.01, *p<0.05, n.s. p>0.05. See *Appendix 1—tables 20–29* for detailed statistics. Source data available at 10.5061/dryad.2547d7x15.

The online version of this article includes the following figure supplement(s) for figure 5:

**Figure supplement 1.** Analysis of population-level odor encoding.

---

*supplement 7B*, *Appendix 1—table 37*). When compared across days, we found that the percentage of VP neurons that respond to both S odors increases from 6.1 ± 2.2% on day 1–34.1 ± 5.1% by the 6th day of imaging (*Figure 3E*,*Appendix 1—table 11*). By comparison, the percentage of OT neurons that respond to both S odors in the same direction (i.e. excited by both S odors or inhibited by both S odors) did not increase through training. Furthermore, whereas $OT_{D1}$ and $OT_{D2}$ neurons were more likely to respond to a single odor than they were to respond to both S odors (12.6 vs 31.3% in $OT_{D1}$, 11.8 vs 21.7% in $OT_{D2}$), VP neurons were more likely to respond to both S odors than to a single odor (34.1 vs 23.3%).

Similarly, we found that the magnitude of trial-averaged odor responses in the VP were significantly higher for S odors than X or P odors on the last day of imaging (*Figure 2*, *Figure 2—figure supplement 8*, *Appendix 1—table 38*). By comparison, neither sucrose-pairing nor airpuff-pairing had any impact on the magnitude of odor responses in $OT_{D2}$ neurons on day 6. And though we did observe a significant effect of sucrose-pairing on response magnitudes in $OT_{D1}$ neurons, both the effect size and significance were weaker than observed in VP. We propose that an ideal valence-encoding neuron should respond similarly to two odors of equal reward-contingency but disparate molecular structure, and we looked at the correlation between each neuron's response to the sucrose-paired ketone ($S_K$) and to the sucrose-paired terpene ($S_T$). VP neurons had a high correlation between a neuron's responses to $S_K$ and $S_T$ (*Figure 3F*; $R^2$=0.89). This similarity was much higher than between the sucrose-paired ketone ($S_K$) and the control ketone ($X_K$) despite the greater structural similarity between $S_K$ and $X_K$ (*Figure 3G*; $R^2$=0.33). In contrast, for both $OT_{D1}$ and $OT_{D2}$ neurons, there was a higher correlation between responses to similar molecular structure ($S_k$ and $X_K$) than between responses to similar contingency ($S_K$ and $S_T$) ($OT_{D2}$: $S_K$ vs. $S_T$ $R^2$=0.04, $S_K$ vs. $X_K$ $R^2$=0.58; $OT_{D1}$: $S_K$ vs. $S_T$ $R^2$=0.13, $S_K$ vs. $X_K$ $R^2$=0.40). Moreover, most VP neurons (76.5%), had a smaller absolute difference in the response magnitude to the 2 S odors ($|S_K-S_T|$) than the absolute difference between the sucrose-paired ketone and the control ketone ($|S_K-X_K|$) (*Figure 3H*). By comparison, only half of $OT_{D2}$ and $OT_{D1}$ neurons showed smaller $|S_K-S_T|$ than $|S_K-X_K|$, as would be expected if response magnitude to an odor did not depend on reward-contingency. This trend was not due to the fact that VP neurons were more likely to respond to both S odors than the OT neurons were since it was consistent across various thresholds for odor response magnitude. This trend was consistent for other pairwise odor comparisons where one odor was a sucrose-cue and the other was not (e.g. $S_K$ vs. $P_T$, *Figure 3—figure supplement 1A, B*).

Finally, we reasoned that the activity of reward-contingency encoding neurons would support good decoding of odor pairs which have different valence but not of odor pairs that have the same valence. To do this, we trained binary logistic classifiers from each neuron's response to all 15 odor pairs and quantified the area under their receiver operating characteristic (auROC). Because auROC values were non-normal with large spread, we quantified what percentage of neurons had an auROC of at least 0.75, halfway between ideal and at-chance decoding. We also note that all classifiers with auROC >0.75 showed bootstrapped p-values less than $10^{-3}$ (*Figure 3—figure supplement 1C and D*). To assess whether neurons from each region were encoding valence, we compared a neuron's {$S_K$ vs. $X_K$} decoder performance (*inter*valence classification) against its {$S_K$ vs. $S_T$} decoder performance (*intra*valence classification) (*Figure 3I–K*, *Figure 3—figure supplement 1E and F*). Across multiple days of imaging, we found that the percentage of neurons that support intervalence classification increased regardless of region but that this effect was markedly more pronounced among VP neurons than among $OT_{D1}$ or $OT_{D2}$ neurons (*Figure 3I–J*, *Appendix 1—tables 12–15*, *Figure 3—figure supplement 1F*,*Appendix 1—tables 39–41*). Intravalence classification, however, did not depend on days or region (*Figure 3K*,*Appendix 1—tables 16 and 17*, *Figure 3—figure supplement 1F*, *Appendix 1—tables 42–44*). By day 6, there were thrice as many VP neurons with good intervalence

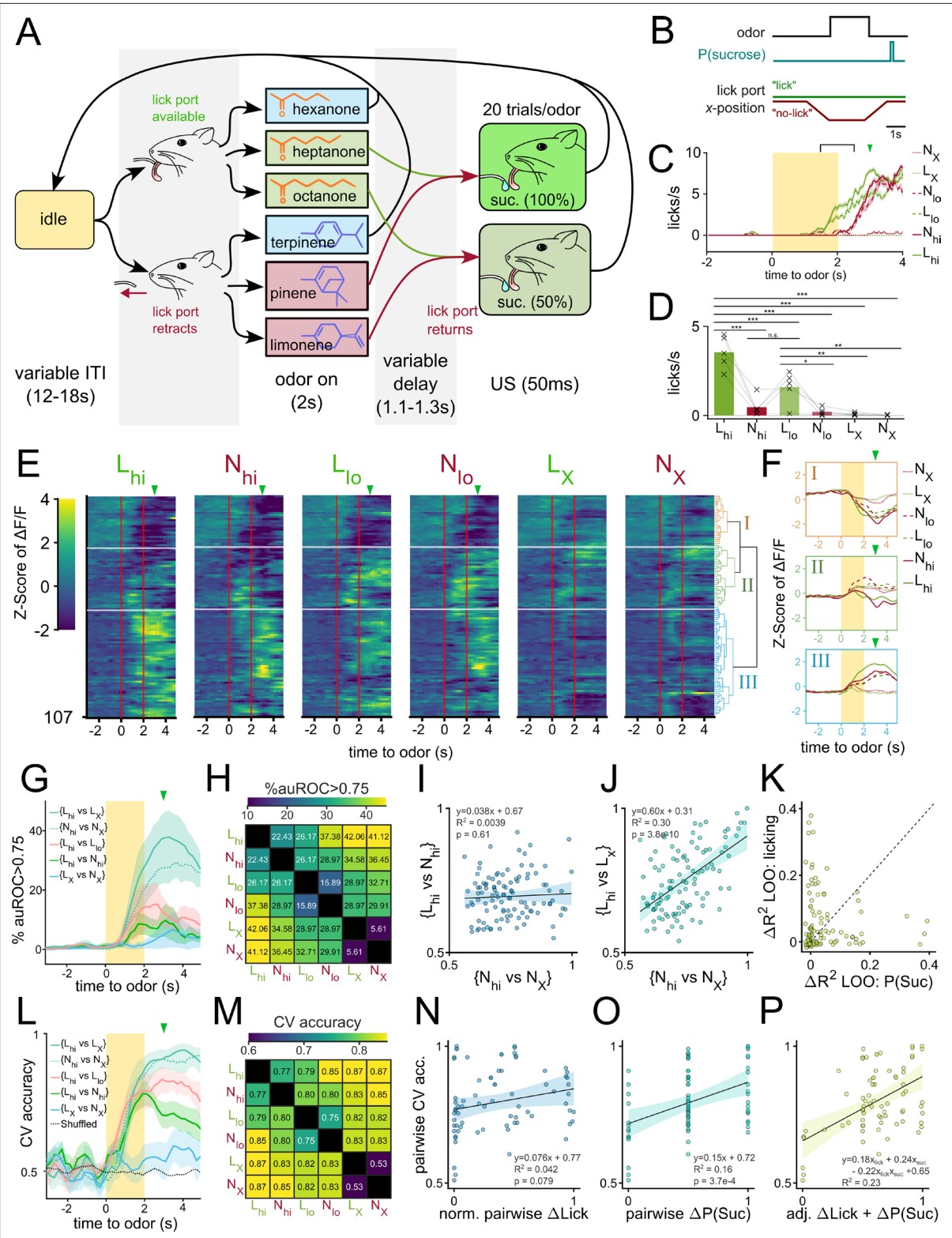

**Figure 6.** Separate VP populations encode reward-contingency and licking vigor. (**A**) State diagram for odor pairing paradigm where lick spout is removed during the presentation of half of the odors. The paradigm is similar to one described in **Figure 2A** with the following key differences: (1) the lick spout is moved away from the animal's mouth during the presentation of half of the odors ($N_{hi}$, $N_{lo}$, $N_X$). (2) sucrose is delivered after a longer variable delay (1.1–1.3 s). (3) 2 of the odors have 100% sucrose contingency ($L_{hi}$, $N_{hi}$), 2 of the odors have 50% sucrose contingency ($L_{lo}$, $N_{lo}$), and the other 2 have 0% sucrose contingency ($L_X$, $N_X$). (**B**) Schematic showing the timing of lick port movement relative to odor and sucrose delivery. (**C**) Licking behavior to

*Figure 6 continued on next page*

*Figure 6 continued*

6 odors averaged across 30 trials from a representative animal. Duration of odor delivery is marked by the shaded rectangle and the average time of sucrose delivery is marked by the arrowhead. The time bin used for subsequent analysis (last 0.5 s of odor and first 0.5 s of delay) is outlined by square brackets (**D**) Average licks/s for each odor measured between the last 0.5 s of odor and the first 0.5 s of delay. Data were pooled from the day of highest difference between licks to $L_{hi}$ and $N_{hi}$. (**E**) Heatmap of odor-evoked activity in VP neurons pooled from each animal's day of highest difference between licks to $L_{hi}$ and $N_{hi}$. Neurons are grouped according to the clustering dendrogram, shown on the right. Horizontal white lines demarcate the boundaries between the three clusters. Odor delivery is marked by vertical red lines. (**F**) Average Z-scored activity of each cluster to each of the six odors. Yellow bar indicates 2 s of odor exposure. (**G**) The percentage of single-neuron linear classifiers with auROC >0.75 as a function of time relative to odor delivery. Shaded area represents the SEM across biological replicates (n=5). (**H**) Heatmap of the percentage of pooled VP neurons with auROC >0.75 during the last 0.5 s of odor and first 0.5 s of delay. (**I**) Scatterplot comparing the auROC for {$L_{hi}$ vs $N_{hi}$} (y-axis) and {$N_{hi}$ vs. $N_x$} (x-axis) for each neuron. The line of best fit is plotted as a dotted line, with the 95% confidence interval shaded in. (**J**) Same as (**I**) but comparing the auROC for {$L_{hi}$ vs $L_x$} (y-axis) and {$N_{hi}$ vs. $N_x$} (x-axis). (**K**) Scatterplot comparing regression models that explain each neuron's activity on a given trial as a function of anticipatory licking or sucrose contingency. The values plotted are the loss in $R^2$ in models without anticipatory licking (y-axis) or sucrose contingency (x-axis) when compared to a model with both variables and their interaction term. (**L**) CV accuracy for five different odor pairs as a function of time relative to odor delivery. (**M**) Heatmap of average pairwise CV accuracy trained on the last 0.5 s of odor and the first 0.5 s of delay. (**N**) Scatterplot of all pairwise classifier accuracies from all animals (y-axis) and the corresponding range-normalized average pairwise difference in anticipatory licking (x-axis). (**O**) Scatterplot of all pairwise classifier accuracies from all animals (y-axis) and the corresponding pairwise difference in reward-contingency (x-axis). (**P**) Scatterplot of all pairwise classifier accuracies (y-axis) and the adjusted combined model of ranged-normalized Δlick and Δreward-contingency (x-axis). FWER-adjusted statistical significance for post hoc comparisons are shown as: \*\*\*p<0.001, \*\*p<0.01, \*p<0.05, n.s. p>0.05. See *Appendix 1—tables 30 and 31* for detailed statistics. Source data available at 10.5061/dryad.2547d7x15.

The online version of this article includes the following figure supplement(s) for figure 6:

**Figure supplement 1.** Camera-based detection of licking in head-fixed animals.

decoding than with intravalence decoding (51.8 ± 5.0% vs 14.4 ± 5.8% for {$S_K$ vs $X_K$} and {$S_K$ vs $S_T$}, respectively). In contrast, a similar number of OT neurons displayed good intervalence decoding as did intravalence decoding (20.8% vs 19.9% of $OT_{D1}$; 12.8% and 21.0% of $OT_{D2}$ for {$S_K$ vs $X_K$} and {$S_K$ vs $S_T$}, respectively). The pattern of better intervalence decoding than intravalence decoding among VP neurons was observed across all 15 pairwise classifiers (*Figure 3—figure supplement 1H*). Whereas 10.2% of all day 6 VP neurons had auROC >0.75 for {$S_K$ vs. $S_T$}, 46.9–57.8% had auROC >0.75 for any classification between a sucrose-cue and a control odor or airpuff-cue. By comparison, there were few neurons with auROC >0.75 for any classification between a puff-cue and a control odor (2.3–10.9%), suggesting that negative valence is either not encoded in these VP neurons or the negative valence was not learned.

Plotting a neuron's {$S_K$ vs. $S_T$} auROC against its {$S_K$ vs. $X_K$} auROC, we can categorize a neuron into the 4 categories (*Figure 3L*). (1) a valence encoding neuron ({$S_K$ vs. $S_T$}<0.75 and {$S_K$ vs. $X_K$}>0.75), (2) an identity encoding neuron (both auROC >0.75), (3) an identity encoding neuron that does better with S odors ({$S_K$ vs. $S_T$}>0.75 and {$S_K$ vs. $X_K$}<0.75), and (4) an uninformative neuron (both auROC <0.75). According to this categorization, half of VP neurons were valence encoding by day 6, followed by $OT_{D1}$ then $OT_{D2}$ (*Figure 3M and N*; 47.7, 16.2, 7.3% for VP, $OT_{D1}$, and $OT_{D2}$, respectively). The opposite was true for identity encoding. VP had a smaller percentage of identity encoding neurons than either $OT_{D1}$ or $OT_{D2}$ (14.8, 21.1, 22.9% for VP, $OT_{D1}$, and $OT_{D2}$, respectively). We note that these conclusions can also be replicated when analyzing multinomial regression (MNR) classifiers trained on single neuron activities *Figure 3—figure supplement 2F and G*, *Appendix 1—tables 50–53*. Namely, the rates of confusion between the 2 sucrose cues are highest in VP and lowest in $OT_{D2}$ whereas the rates of confusion across all ketones ($S_K$, $X_K$, $P_K$) are highest in $OT_{D2}$ and lowest in VP. These single-neuron classifier analyses further indicate that VP neurons, more than either $OT_{D2}$ or $OT_{D1}$ neurons, were encoding reward contingency at the single neuron level. However, the most striking observation was that while only a subset (37.5%) of VP neurons had auROC <0.75 for both {$S_K$ vs. $X_K$} and {$S_K$ vs. $X_K$}, a majority of $OT_{D2}$ and $OT_{D1}$ neurons (69.7% and 62.6%, respectively) showed auROC <0.75 for both {$S_K$ vs. $S_T$} or {$S_K$ vs. $P_K$}. Thus, in comparison to the VP, most individual anteromedial OT neurons have little discriminatory information about olfactory stimuli regardless of valence at the single-neuron level and may be better suited in a population code.

Our data indicated that valence encoding emerges in VP neurons over the course of learning. To explore the potential mechanisms at the cellular level, we compared the activity of a subset of neurons we could observe on both day 1 and day 3 (*Figure 4A–F*). We noticed there were neurons that responded to the sucrose delivery on day 1 that responded to the sucrose cue on day 3 (*Figure 4C*

*and F*), reminiscent of models of Hebbian plasticity. When quantified, we found that 17.9, 20.9% of VP neurons were responsive to sucrose on day 1 and $S_K$ and $S_T$ on day 3, respectively (*Figure 4G*). We specifically considered neurons that had the same direction of response (excitation or inhibition) to both cues on separate days. This figure was much lower among OT subpopulations (11.5, 8.2% for $S_K$ and $S_T$ in $OT_{D1}$; 10, 2.5% for $S_K$ and $S_T$ in $OT_{D2}$). Consistent with above observations, we also found that the odor responses to sucrose-cues were larger on day 3 than day 1 in 85% of tracked VP neurons, but only in 65% and 57% of $OT_{D1}$ and $OT_{D2}$ neurons, respectively (*Figure 4H–I*, *Appendix 1—tables 18 and 19*). We did not see the same effect in VP neurons' responses to control or puff-paired odors. Together, our data suggest that sucrose pairing causes sucrose-responsive VP neurons to increase their responses to the sucrose-predictive odors.

Olfactory brain areas are known to use population codes to encode sensory information, whereby single neurons have weak discriminatory information, but the activity of the population allows for an efficient encoding of high-dimensional data. To assess if there is discriminatory information about the odorants within the population-level activity, we compared the pairwise Euclidean distance of trial-averaged odor responses for all 15 odor pairs (*Figure 5A and B*). We saw that, in general, the pairwise Euclidean distance for all odor pairs examined increases quickly after the onset of odor, reaches peak distance towards the end of the 2 s odor delivery, and slowly decays after odor ends (*Figure 5A*). When examining the average pairwise distance during the last second of odor, there was a relatively unstructured distribution of pairwise distance in $OT_{D2}$ odor-response such that $\|S_K-X_K\|$, $\|S_K-P_K\|$, $\|S_K-S_T\|$, and $\|X_K-X_T\|$ were all similar (*Figure 5B*). By comparison, in VP populations, the distribution was structured such that intervalence pairwise comparisons between sucrose-paired and not sucrose-paired odors (e.g. $\|S_K-P_K\|$ and $\|S_K-X_K\|$) were larger than intravalence pairwise comparisons (e.g. $\|S_K-S_T\|$, or $\|X_K-X_T\|$). $OT_{D1}$ populations showed an intermediate trend where most intravalence pairwise distances were smaller than intervalence pairwise distances with the exception of $\|S_K-S_T\|$. Thus, at the population level VP representations appear to encode valence but not identity, whereas the anteromedial OT representations encode some valence information but appear to be better suited for identity encoding.

In parallel, we also performed decoding analysis using linear classifiers to assess how reliably a given pair of odors could be decoded from population-level activity (*Figure 5C–D*). To quantify this, we extracted the average $\Delta F_{i,k}/F$ values for each trial $i \in [1,m]$ and each neuron $k \in [1,n]$. The resulting matrix of size $m \times n$ was used to train a binary linear classifier with a logistic learner. For each classifier, we looked at the average accuracy across fivefold cross-validation (CV accuracy). Classifiers were trained on simultaneously recorded populations (i.e. neurons from the same animal recorded on the same day) to capture biological variability. A total of 765 pairwise linear classifiers were trained (15 pairwise comparisons, 17 animals, and 3 days). When compared against 10,000 shuffles, 569 of these classifiers showed bootstrapped p-value less than 0.001 (*Figure 5—figure supplement 1A*). Importantly, all classifiers with CV accuracy higher than 0.75 had p-value less than 0.001.

Linear classifiers trained on day 6 $OT_{D2}$ population data had similar ranges of accuracy regardless of valence (*Figure 5D*). For example, the intravalence classification {$S_K$ vs. $S_T$} was more accurate (86.6 ± 3.9%) than some and intervalence classifications (e.g. {$S_K$ vs. $X_K$}, 72.8 ± 5.4%) but less accurate than others (e.g. {$S_T$ vs. $P_K$}, 88.2 ± 3.1%). Classifiers trained on VP population activity, however, always showed more accurate intervalence decoding (range: 89.5–96.1%) than intravalence decoding ({$S_K$ vs. $S_T$}, 79.9 ± 6.1%). Additionally, whereas $OT_{D2}$ population classifiers could decode the 2 control cues {$X_K$ vs. $X_T$} at accuracy (85.8 ± 4.2%) comparable to sucrose-cue vs. non-sucrose-cue, VP population classifiers were consistently less accurate (76.8 ± 3.7%) at {$X_K$ vs. $X_T$} than the aforementioned intervalence classifiers. This suggests that whereas $OT_{D2}$ encodes odor identity agnostic to the valence, VP does not encode identity at all but rather encodes reward contingency or positive valence. $OT_{D1}$ pairwise classification was a mixture of the other 2 regions: sucrose-cue vs. non-sucrose-cue classification was more accurate than most other pairwise classifications (range: 86.4–94.3%), but the {$S_K$ vs. $S_T$} classification was comparably accurate (90.9 ± 4.7%). This rules out the interpretation that $OT_{D1}$ strictly encodes valence since the identity of 2 sucrose-cues can be decoded well.

To address the possibility that our results are due to the limitations of linear classification, we repeated the analysis using support vector machines (SVMs) with a radial basis function kernel and found we could draw the same conclusions (*Figure 5—figure supplement 1E*). Similarly, to verify our results are not epiphenomena of forcing the data into binary classification, we looked

at population-level MNR classifiers trained on day 6 data. Importantly, we observe high confusion between 2 sucrose cues in MNR classifiers trained on VP data, but not those trained on $OT_{D2}$ or $OT_{D1}$ data (*Figure 5—figure supplement 1F*), corroborating through an alternate analysis method that VP population activity encodes reward contingency whereas either anteromedial OT subpopulations are better at encoding identity.

The fact that VP populations showed higher decoding for odor pairs of unequal sucrose-contingency provides strong evidence that VP encodes reward-contingency more than identity. Results from OT decoder analyses, however, are less intelligible: all 15 odor pairs, regardless of sucrose-contingency, could be decoded with above-chance success. Although this result is consistent with OT populations encoding identity rather than valence, it does not rule out the possibility that valence and identity are both encoded. In the context of cue-association, two cues of different valence cannot have the same identity, meaning that good decoding of $\{S_K$ vs. $P_K\}$ can be extracted from either valence encoding or identity encoding populations. To disambiguate these two possibilities, we looked at the generalizability of pairwise decoders. Briefly, linear classifiers were trained on each of the 15 possible odor pairs. Afterwards, the resulting classifier was tested on every other odor pair (*Figure 5E*). We reasoned that if neural populations encode valence in addition to identity, classifiers trained on any odor pair of unequal sucrose-contingency should consistently perform above chance on a different odor pair of unequal sucrose-contingency (e.g. train on $\{S_K$ vs. $P_K\}$, test on $\{S_T$ vs. $X_T\}$). In other words, given valence encoding, $\{S_K$ vs. $P_K\}$ should be discriminable in a way that can also discriminate $\{S_T$ vs. $X_T\}$. As expected, VP population decoders were consistently generalizable when trained on odor pairs of unequal sucrose-contingency then tested on other odor pairs of unequal sucrose-contingency (*Figure 5F and G*). $OT_{D2}$ population decoders, on the other hand, showed negligible generalizability across pairs of unequal sucrose-contingency. Similarly to other metrics of valence encoding, we found that $OT_{D1}$ displayed a generalizability in between that of VP and $OT_{D1}$, suggesting that $OT_{D1}$ could encode some valence in addition to identity. However, we note that the VP population, on average, outperforms $OT_{D1}$ at generalized valence decoding (95.0 ± 2.0% vs 78.5±3.9%; *Appendix 1—tables 22 and 23*).

After performing these population-level analyses, we noticed a discrepancy: although single-neuron intervalence decoding was worse in anteromedial OT than in VP (*Figure 3M–N*), population-level intervalence decoding was comparable between either OT subpopulations and the VP (*Figure 5C–D*). This led us to speculate that the encoding of odor information had a higher dimensionality in OT than in VP. To explicitly compare the dimensionality of VP and OT population activities, we looked at the extent to which the population vector is spread across multiple axes using principal component analysis (PCA). Dimensionality can further be quantified using the participation ratio (PR) of a population, which is the square of the sum of eigenvalues of its covariance matrix divided by the sum of the squares of its eigenvalues (*Litwin-Kumar et al., 2017*; *Recanatesi et al., 2019*). This value will have a range of 1 to n, where n is the total number of features. If a single principal component can describe all of the total population variance (i.e. the data is low-dimensional), the population will have PR equal to 1. Conversely, if every principal component equally describes $n^{th}$ of the total variance (i.e. the data is high-dimensional), the population will have PR equal to n. Because the number of total neurons recorded was different between OT and VP experiments, we first assessed if and how the normalized PR would vary with the number of total neurons through random sampling (*Figure 5H*). After observing a consistent decrease in PR with increasing n, we compared the PR of OT and VP animals by repeatedly subsampling a fixed number of neurons (k=15) and found that VP animals had lower PR ($PR_{VP}$ = 5.83 ± 0.80) than either $OT_{D2}$ ($PR_{D2}$=9.61 ± 0.37) or $OT_{D1}$ ($PR_{D1}$=9.24 ± 0.44) animals after training (*Figure 5I*, *Appendix 1—tables 24 and 25*). There was also a difference, however, in how valence information vs. identity information was encoded by VP populations. Though the first PC of each VP population was sufficient to train adequate $\{S_K$ vs. $P_K\}$ decoders (CV accuracy$_{PC1}$=85.5 ± 2.7%), all 15 PCs were required for comparable $\{S_K$ vs. $S_T\}$ decoding (CV accuracy$_{PC1:15}$=75.1 ± 13.4%) (*Figure 5J–L*, *Appendix 1—tables 26–29*). In either OT populations, the first PC did not support good decoding of either $\{S_K$ vs. $P_K\}$ or $\{S_K$ vs. $S_T\}$. Together, our population-level analysis indicates that VP encodes valence, but not identity, in low-dimensional space, $OT_{D2}$ encodes identity but not valence in high-dimensional space, and $OT_{D1}$, has some valence information and encodes identity in high-dimensional space.

Analyses at the single-neuron and population levels showed that VP activity encodes reward contingency, rather than the identity, of the olfactory stimulus. However, due to the task design, the reward-contingency of a stimulus was highly correlated with the vigor of licking (*Figure 2F*). This raised concerns that some neurons classified as robust reward-contingency encoders were potentially encoding motor-related information. Indeed, many VP neurons showed consistent increases in fluorescence time-locked to the onset of a licking-bout (*Figure 6—figure supplement 1A, B*), and could be used to train distributed lag models to predict onset of licking bouts (*Figure 6—figure supplement 1C*). Across all VP neurons, we observed a positive and significant correlation between a neuron's valence decoding ability and licking decoding ability (*Figure 6—figure supplement 1D*; slope = 0.41, p=$2.2 \times 10^{-10}$, $R^2$=0.28). This motivated us to develop a new conditioning paradigm that could decouple reward-contingency of an odor cue from the behavioral output.

Initially, we attempted to train animals on a symmetric Go/No-Go operant task where reward delivery was contingent on licking or withholding licks during odor. However, consistent with previous findings (*Gubner et al., 2010*), we found that animals struggled to learn the No-Go behavior in comparison to the Go behavior (data not shown). In an operant paradigm, this leads to a problematic difference in valence of Go/No-Go cues. Consequently, we opted to develop a classical conditioning paradigm whereby licks were encouraged/discouraged by physically moving the lick spout before odor presentation (*Figure 6*).

Briefly, headfixed animals were presented with one of six odors in pseudorandomized order. During the presentation of three of these odors, the lick spout was moved away from the mouse with a linear stepper motor. These odors are denoted as N odors (N for No-lick spout). During the presentation of the other three odors, the lick spout remained within licking distance of the mouse's tongue. These odors are referred to as L odors (L for lick spout). One odor from each group served as a control odor that had 0% reward-contingency ($L_x$, $N_x$). The other two odors in each group were paired with sucrose at low (50%) or high (100%) probability ($L_{lo}$, $L_{hi}$, $N_{lo}$, $N_{hi}$). We reasoned that this contingency could allow us to make pairwise comparisons where one odor has a higher value but lower anticipatory licking than the other (e.g. $N_{hi}$ vs. $L_{lo}$).

To monitor anticipatory licking in the absence of the lick spout, we trained a distributed lag model (DLM) using features of the mouse's face tracked using DeepLabcut (*Mathis et al., 2018*; *Figure 6—figure supplement 1E–G*). We chose to pool data across all mice from the day of the highest licking differential between $L_{hi}$ and $N_{hi}$ odors (*Figure 6—figure supplement 1H*) to maximize the decoupling of value and motor output in our analysis. Anticipatory licking during $L_{lo}$ or $L_{hi}$ began during the last second of the odor and increased gradually until sucrose delivery whereas licking during $N_{lo}$ or $N_{hi}$ was delayed by about one second (*Figure 6C*). When quantified across animals on their days of highest lick differential, we found that mice consistently licked most during $L_{hi}$, followed by $L_{lo}$ and $N_{hi}$, then $N_{lo}$ (*Figure 6D*, *Appendix 1—tables 30 and 31*). Mice showed little to no licking during either control odors. Thus, this behavioral assay affords us the opportunity to assess the decoupled effects of reward-contingency and licking vigor on neural activity.

To begin to characterize the presence of reward-contingency and/or licking vigor encoding in the VP, we first pooled and clustered the neural activity taken from five animals on their days of highest lick-differential (*Figure 6E*). When clustering VP neurons into three clusters, we found that one cluster (I) showed a largely similar inhibitory response to the 4 sucrose-paired odors, but not control odors-much like cluster (I) from the previous conditioning experiment (*Figure 3A–B*, *Figure 6E–F*). Another cluster (III) by comparison, showed a varied excitatory response to each of the four sucrose-paired odors, much like cluster III from the previous experiment. Cluster (III) neurons seemed to have a particularly strong response to $L_{hi}$ for which there was most anticipatory licking. This led us to speculate the existence of both reward-contingency encoding and vigor encoding neurons in the VP.

To test this directly, we quantified single neuron decodability of odor pairs and examined how correlated decoding along the reward-contingency axis is to decoding along vigor axis (*Figure 6G–H*). We reasoned that auROC values for {$L_{hi}$ vs. $N_{hi}$} would be high for vigor encoding neurons but not value encoding neurons given these two odors have the same reward-contingency but disparate licking behaviors. Similarly, we reasoned that auROC values for {$N_{hi}$ vs. $N_x$} would be high for reward-contingency encoding neurons but not vigor encoding neurons given there is a large difference in value but small difference in licking between these two odors. First, we saw that while single neuron decodability along the reward-contingency axis (e.g. {$N_{hi}$ vs $N_x$}) was higher than along the lick axis

(e.g. {$L_{hi}$ vs $N_{hi}$}), there were more neurons that could decode {$N_{hi}$ vs $N_X$} than could decode {$L_X$ vs $N_X$} at auROC >0.75 (**Figure 6G–H**). Furthermore, we saw a lack of significant correlation between the single-neuron decodability of two odors that had similar licking but different reward-contingency ({$N_{hi}$ vs $N_X$}) and the decodability of two odors that had different licking but same reward-contingency ({$L_{hi}$ vs. $N_{hi}$}) (**Figure 6I**; slope=0.038, p=0.61, $R^2$=0.0039). This decoupling suggests that reward-contingency and vigor information are both encoded in the VP but by different populations. As a control, we saw a significant correlation between two pairwise comparisons that both had high difference in reward-contingency ({$N_{hi}$ vs $N_X$} and {$L_{hi}$ vs $L_X$}) (**Figure 6J**; slope = 0.60, p=3.8 × 10$^{-10}$, $R^2$=0.30).

We also performed the converse experiment where the $\Delta\Delta F/F_{baseline}$ values of each neuron were linearly fitted to either (1) the reward contingency, (2) the anticipatory licking or (3) both values and the interaction term. Then, we compared the $\Delta R^2$ when either variable was omitted in the model and plotted the $\Delta R^2_{-valence}$ against the $\Delta R^2_{-licking}$ (**Figure 6K**). We reasoned that, if a typical VP neuron's activity could be well-explained by either reward-contingency or vigor but not both, we would see points along either x or y-intercepts. On the other hand, if a typical neuron's activity could be well-explained by a linear combination of the two variables, we would see data fall along a line of positive slope. We found that most neurons tended to have large $\Delta R^2_{-valence}$ *or* large $\Delta R^2_{-licking}$ values but not both, supporting the idea that two largely non-overlapping sets of VP neurons encode reward-contingency *or* vigor but not both.

Lastly, we trained linear classifiers of pairwise odor comparisons using population-level activity to assess if both reward-contingency and vigor information were present in the population-level activity. Consistent with single-neuron decoder analysis, we found that {$L_{hi}$ vs $L_X$} and {$N_{hi}$ vs $N_X$} could both be decoded better than {$L_X$ vs $N_X$} (**Figure 6L, M**). Because we train each classifier using simultaneously recorded neural activity (i.e. from a single animal), we had a total of 75 classifiers (15 pairwise classifiers for 5 animals). The cross validated accuracies of these classifiers were then fitted to a linear model of pairwise differences in either (1) reward-contingency, (2) anticipatory licking, or (3) both. If the population VP activity encodes either reward-contingency or vigor but not both, we expect to see one of the single-variable models outperform the other greatly. But if the population VP activity encodes both variables, we expect the multivariable model would outperform either single model. We found that $\Delta$licking has a weak and not significant relationship with pairwise CV accuracy (**Figure 6N**; slope = 0.076, p=0.079, $R^2$=0.042). By comparison $\Delta$reward-contingency (or P(S), as in probability of sucrose delivery following odor) had a larger and statistically significant correlation with pairwise accuracy (**Figure 6O**; slope = 0.15, p=3.7 × 10$^{-4}$, $R^2$=0.16). The combined model, however, showed larger coefficients and larger $R^2$ than either single variable model, suggesting an additive effect of both features on CV accuracy (**Figure 6P**; accuracy = 0.18$\Delta$lick +0.24$\Delta$P(S) - 0.22$\Delta$lick*$\Delta$P(S)+0.65, $R^2$=0.23). Thus, we conclude that both reward-contingency and licking vigor are encoded in the population-level activity of VP neurons.

## Discussion

Our anatomical investigations demonstrate that the primary output of the anteromedial OT is to the VP, with minimal connections to the VTA. Given its constrained connectivity, we propose that the anteromedial OT to VP circuit is an ideal model system for examining how the encoding of reward cues is transformed across brain circuits. Utilizing comparative longitudinal imaging, we found that VP, but not $OT_{D2}$, robustly encodes the sucrose-contingency of odors. Although our analyses revealed that sucrose-contingency influences odor-evoked responses in $OT_{D1}$ neurons more so than in $OT_{D2}$ neurons, other evidence suggests valence encoding is not the appropriate framework for interpreting $OT_{D1}$ activity. Specifically, information about sucrose-contingency in $OT_{D1}$ resides in a high-dimensional space and generalizes poorly, whereas VP encodes reward-contingency robustly in a low-dimensional and generalizable manner. Thus, we suggest that the changes in anteromedial $OT_{D1}$ activity are more likely to reflect increased contrast of identity or an intermediate encoding of valence that also encodes identity. Finally, using a novel classical conditioning paradigm, we assigned motor-related signals and expected-value signals to non-overlapping VP subpopulations.

Some of our findings were unexpected. For example, we found no evidence that either $OT_{D1}$ or $OT_{D2}$ have significant extrapallidal outputs. This is in contrast to a previous study which reported that $OT_{D1}$ neurons, and to a lesser extent, $OT_{D2}$ neurons, project to the LH and VTA (**Zhang et al., 2017b**). It is possible that other parts of the OT have extrapallidal outputs, as we only performed anterograde

tracing from the anteromedial portion. It is also possible that at least some of the VTA labeling Zhang and colleagues observed from anterograde viral tracing experiments could be due to backflow of the tracer virus in nuclei immediately dorsal to the OT (e.g. AcbSh). As a critical control, we provide evidence that retrograde tracing from VTA robustly labels AcbSh neurons but hardly any neurons from any part of the OT. And the few VTA projecting OT neurons we did observe were restricted to the distal portions of layer III bordering the VP. Consistent with this, quantification of OT afferents is glaringly absent from 2 independent characterizations of brainwide inputs onto VTA (*Beier et al., 2015*; *Faget et al., 2016*). In contrast, OT has been reported to be one of the most prominent inputs to both GAD2 +and Vglut2 +VP neurons (*Stephenson-Jones et al., 2020*). It is difficult, however, to completely rule out the existence of OTmidbrain projections due to the limitations of our experiments: we primarily targeted layer II in the anteromedial portion of the OT for anterograde tracing and only tested the VTA with retrograde tracers. More posterior and/or lateral portions of the OT could have extrapallidal outputs posterior to the VTA. Despite these caveats, the evidence suggests that *Drd1+* neurons in the anteromedial portion of the OT have little extrapallidal projections when compared to the AcbSh.

Although we found little difference in the output patterns of anteromedial $OT_{D1}$ and $OT_{D2}$ neurons, we observed differences in how these two subpopulations encode odor valence. Consistent with a previous report (*Martiros et al., 2022*), we found that $OT_{D1}$ activity, more than $OT_{D2}$ activity, is modulated by reward contingency. For example, $OT_{D1}$ neurons, but not $OT_{D2}$ neurons, were more likely to respond to sucrose-paired odors than other odors. And the magnitude of responses in $OT_{D1}$ but not $OT_{D2}$ neurons were significantly larger to sucrose-paired odors than to other odors. We refrain, however, from concluding that the primary feature encoded in $OT_{D1}$ neurons is valence or reward contingency, for the following reasons. First, the above-mentioned effects of sucrose-contingency on neural activity are much stronger for VP than for $OT_{D1}$. Additionally, whereas more than 50% of VP neurons could be categorized as reward-contingency encoders, this figure was less than 20% for $OT_{D1}$. Lastly, population-level decoders trained on odor pairs of different valence can generalize in the case of VP populations, but not $OT_{D1}$ populations. While we acknowledge that there is poor standardization when it comes to defining valence encoding, it is unlikely that discrepancies between our conclusions and those of Martiros et al. stem from differences in interpretation alone. Comparative examination of our analyses reveals clear dissimilarities in the effect-size of shared metrics (e.g. % odor responsive). Given the high Z-resolution afforded by 2-photon microscopy, it is probable that we recorded from different layers of the OT, which should not be assumed to have identical physiology. We note that the lens placements in our experiments are considerably more ventral than those reported in Martiros et al. It is possible that these neurons are recorded from layer III of the OT, whereas the majority of the neurons in the present study are recorded from layer II. A direct comparison of layer II and layer III OT neurons and their valence encoding could prove useful in understanding the discrepancies between the two studies. It is also possible that some of the neurons recorded in Martiros et al. could be from the rostral portion of the VP which lies immediately dorsal to layer III of the OT. Although *Adora2a* and *Drd1* are not expressed as mRNA in the VP, the BAC-transgenic lines used for both the present work and work by Martiros et al. label neurons in the VP.

Our comparison of OT and VP is reminiscent of previous comparisons made between value encoding in VP and NAc (*Ottenheimer et al., 2018*; *Richard et al., 2016*). These publications showed that VP encodes incentive value more robustly than the NAc. Given that OT and NAc share many anatomical, physiological, and molecular traits, it is tempting to speculate that the encoding schemes, too, would be similar between the two areas. Optogenetic activation of $OT_{D1}$ supports RTPP (*Murata et al., 2019*), as does activation of D1 or D2 neurons of the NAc (*Soares-Cunha et al., 2020*). While we acknowledge stimulation experiments provide unique insights that cannot be obtained from recordings alone, we note that SPN's have extensive inhibitory collaterals and exhibit high-dimensional activity. Given these peculiarities of the striatum, we predict that bulk stimulation leads to activity patterns well outside the physiologically relevant range and that this warrants conservative extrapolations regarding OT SPN's endogenous role.

An interesting conclusion from our work is that, within the context of our conditioning paradigm, the dimensionality of neural activity was much lower in VP than in OT. Furthermore, the dimensionality of the imaged subpopulations were anti-correlated with the robustness of sucrose-contingency encoding: $OT_{D2}$ displayed the highest dimensionality and lowest valence encoding whereas VP

displayed the lowest dimensionality and highest valence encoding. As discussed elegantly by others (*Chu et al., 2016*; *Shannon, 1948*), there is generally a tradeoff between the efficiency of a neural population (i.e. its total information capacity) and the robustness of its encoding scheme (i.e. redundancy of encoding). Consistently, it is likely that VP neurons display such robust encoding of valence, in large part, due to the loss of odor identity information. By comparison, OT populations may be able to encode information about the large olfactory identity space due to their high dimensionality. We speculate that the extensive inhibitory collaterals among SPN's play a role in enforcing the high dimensionality of OT activity. Though it is entirely unknown what anatomical or physiological strategies are used to reduce VP dimensionality, we consider this an important piece of the puzzle in understanding VP computations.

We saw little evidence of negative valence neurons in any of the 3 populations that were imaged. This was surprising given previous reports of negative valence neurons in the VP (*Stephenson-Jones et al., 2020*). We consider two potential explanations for this discrepancy. First, it is possible that our conditioning paradigm was not sufficiently aversive for the animals. Although our behavioral evidence for aversive association is significant, it is less robust than sucrose association raising the possibility that the learning was insufficient. This could be due to the fact that we targeted the airpuff to the animal's hindquarters rather than to the face. But we note that in a previous report, airpuff delivery to the snout and to hindquarters elicited similar ingress response in a burrowing assay (*Fink et al., 2019*). Additionally, we observed clear unconditioned responses to the airpuff itself. Another possibility is that, while negative valence neurons do exist in the VP, as has been reported, they were outside of our field-of-view. Previous work in the VP supports positive and negative valence as being encoded by *Vgat+* and *Vglut2+* neurons, respectively (*Faget et al., 2018*; *Stephenson-Jones et al., 2020*). Most *Vglut2+* neurons are found in the dorsomedial portion of the VP, whereas our lenses were specifically targeted to the ventrolateral portion where we found the most OT afferents. Given this distinction, our results are not inconsistent with previous reports of negative valence neurons in the VP.

In this work, we present evidence that may appear to contradict previous anatomical and physiological characterizations of the OT. We find that the anteromedial portion of the OT sends high-dimensional information about odor identity primarily to the VP and not the VTA. By directly comparing OT and VP population-level activity in the same paradigm, we bridge together, for the first time, the fields of OT and VP. This provides valuable context which not only helps us evaluate past conclusions about valence encoding in the OT but also consider the implications of the stimulus-evoked activity in the OT. This comparative approach leads us to conclude that the anteromedial OT has relatively little valence information. However, our findings are not generally inconsistent with what has been observed in previous studies. We do find reward modulation in the $OT_{D1}$ population, however, we do not find valence encoding single neurons and the population vector does not generalize between two rewarded odors as it does in the VP. Therefore, we propose that representation in the anteromedial OT reflects either an intermediate representation of reward-contingency or a contrast modulation to reflect the contingency.

## Speculation

It is interesting to note the discrepancy between the anatomical organization of dorsal striatum (DS) vs. ventral striatum (VS): SPN's of the DS project exclusively to either the substantia nigra pars reticulata (SNr) of the midbrain (Drd1+) or the exterior portion of the globus pallidus (GPe) (Drd2+), but Drd1 +neurons in the VS (Acb) project to both the VTA of the midbrain and the ventral pallidum (*Kupchik et al., 2015*). The anteromedial OT appears to have further limited output divergence, whereby both OTD1 and OTD2 neurons project primarily to the VP. This may reflect at a gradient of anatomical connectivity where the most dorsal Drd1 +SPN's project primarily to the midbrain and the most ventral Drd1 +SPN's (i.e. OTD1 neurons) project primarily to the pallidum. Functionally, the lack of evidence for OTD1 to midbrain connectivity challenges the dichotomy of direct vs. indirect pathways in the ventral basal ganglia. In this model, DA orchestrates motor initiation by oppositely modulating Drd1 +and Drd2 +SPN's, which have differential downstream targets. Given the lack of clear differences in OTD1 and OTD2 projections, we think this canonical model of basal ganglia connectivity inadequately explains the functional consequences of DA modulation in the OT.

In our work, we described key differences in how reward cues are encoded in 2 synaptically connected nuclei. What insights can we infer about the role of OT on shaping VP activity through

this comparison? The most salient observation of VP activity is the large and widespread excitatory responses to sucrose-cues. Though the effect size is smaller, $OT_{D1}$ neurons also showed larger excitatory responses to sucrose-cue when compared to other odors. Given that these neurons are GABAergic and their primary target are the VP neurons, it is difficult to explain how these two responses are related. We consider three possible explanations for this paradox. First, in addition to large excitatory responses that were specific to the sucrose-cues, we also observed inhibitory responses that were specific to the sucrose-cues. It is possible that the excitatory VP activity during sucrose-cue presentation is driven mainly by the numerous excitatory afferents (Pir, BLA, etc.) while the inhibitory VP activity is driven mainly by $OT_{D1}$ and $OT_{D2}$ afferents. In a second model, there could be mechanisms downstream of somatic activity that could explain the discrepancy. For example, although brief optical stimulation of D2 neurons in Acb leads to a decrease of VP activity, prolonged activation causes an increase in VP activity via the δ-opioid receptor (*Soares-Cunha et al., 2020*). Our experiments do not provide any information on how neuropeptide release from OT neurons is different during presentation of sucrose-cue vs. control odor. Similarly, we cannot measure if and how positively valent stimuli change the input-output-function of OT neurons. Previous reports have found that Drd2 agonism in Acb neurons leads to a decrease in collateral inhibition through a presynaptic mechanism (*Dobbs et al., 2016*). Given that more DA is expected to be released during presentation of sucrose-cues, it is plausible that the probability of GABA release from OT boutons onto VP dendrites is affected. In a third and perhaps the most parsimonious model, endogenous OT activity does not contribute significantly to explaining the bulk excitatory activity in VP. This goes against the prevailing working model in Acb to VP circuit which assumes that Acb excitation leads to VTA disinhibition by inhibiting the VP. And while there is evidence supporting from bulk stimulation of D1 or D2 neurons in the Acb (*Soares-Cunha et al., 2020*), under endogenous conditions, both Acb neurons and VP neurons are excited in response to reward-cues (*Lederman et al., 2021*; *Ottenheimer et al., 2018*). Furthermore, given that GABAergic synapses from SPN's to VP neurons is likely dendritic (*Bolam et al., 1986*), we think it is unlikely that OT to VP drives large-scale shunting of action potential in the presence of excitatory drive from other areas known to respond preferentially to reward cues such as the BLA (*Beyeler et al., 2018*) or the OFC (*Wang et al., 2020*). We consequently propose an alternate framework in which the mechanistic role of the OT in this circuit is to provide spatiotemporally precise inhibition to coordinate the integration of excitatory inputs onto VP. This form of inhibition could gate which excitatory synapses go through Hebbian potentiation vs. anti-Hebbian depression. Under such a framework, OT would function as a high-dimensional filter for VP neurons to adaptively scale its various excitatory afferents.

## Methods
### Stereotaxic surgery

All procedures were approved by the UCSD Institutional Animal Care and Use Committee. Animals were anesthetized with isoflurane (3% for induction, 1.5–2.0% afterward) and placed in a stereotaxic frame (Kopf Model 1900). Mouse blood oxygenation, heart rate and breathing were monitored throughout surgery, and body temperature was regulated using a heating pad (Physio Suite, Kent Scientific). A small craniotomy above the injection site was made using standard aseptic technique. Virus was injected with needles pulled from capillary glass (3-000-203-G/X, Drummond Scientific) at a flow rate of 2 nl/s using a micropump (Nanoject III, Drummond Scientific). For OT anterograde tracing experiments, 50 µl of AAV9-phSyn1-FLEX-tdTomato-T2A-SypEGFP-WPRE diluted to $10^{12}$ vg/ml (The Salk Institute GT3 core) was injected into the rostral portion of the medial OT (AP: 1.6 mm, ML: –1.0 mm, DV: –5.375 mm) in *Drd1*-Cre or *Adora2a*-Cre mice. For VP retrograde tracing experiments, 100 nl of Cholera Toxin Subunit B CF 488 A (Biotium) was injected at into the caudal portion of the ventrolateral VP (AP: 0.75 mm, ML: –1.4 mm, DV: –5.4 mm) and 100 nl of Cholera Toxin Subunit B CF 543 (Biotium) was injected into the dorsomedial VP (AP: 0.75 mm, ML: –1.0 mm, DV: –5.35 mm) in C57BL6/J mice. For VTA retrograde tracing experiments, 100 nl of Cholera Toxin Subunit B CF 647 (Biotium) was injected to the rostral portion of the VTA (AP: –3.1 mm, ML: 0.8 mm, DV: –4.5 mm) in C57BL6/J mice. CTB injections were done at 1 mg/ml dilution in PBS. In some cases, tracers were injected bilaterally and each hemisphere was analyzed independently. Following each injection, the injection needle was left at the injection site for 10 min then slowly withdrawn.

For imaging experiments, the skull was prepared with OptiBond XTR primer and adhesive (KaVo Kerr) prior to the craniotomy. After performing a craniotomy 800 µm in diameter centered around the virus injection site, a 27 G blunt needle was used to aspirate 1.5 mm below the brain surface. For OT imaging experiments, 500 µl of AAV9-syn-FLEX-jGCaMP7s-WPRE (Addgene viral prep #104491-AAV9) was diluted to $10^{12}$ vg/ml and injected into the left and rostral portion of the medial OT in D1-Cre or A2A-Cre mice. For VP imaging experiments, 300 µl of AAV9-syn-jGCaMP7s-WPRE (Addgene viral prep #104487-AAV9) was diluted to $10^{12}$ vg/ml and injected into the left and caudal portion of the ventrolateral VP in C57BL6/J mice. Following the viral injection, a head-plate (Model 4, Neurotar) was secured to the mouse's skull using light-curing glue (Tetric Evoflow, Ivoclar Group). At least 30 min after viral injection, a 600 µm GRIN lens (NA,~1.9 pitch, GrinTech) was sterilized with Peridox-RTU then slowly lowered at a rate of 500 µm/min into the craniotomy until it was 200 um dorsal to the injection coordinate. The lens was adhered to the surface of the skull using Tetric Evoflow. We then placed a hollow threaded post (AE825ES, Thorlabs) to act as a housing for the lens and adhered it using Tetric Evoflow. Any part of the skull that was still visible was covered using dental cement (Lang Dental). Finally, the housing was covered with a Nylon cap nut (94922 A325, McMaster-Carr) screwed onto the thread post to protect the lens in between imaging. Animals were left on the heating pad until they fully recovered from anesthesia.

## Histology

Mice were administered ketamine (100 mg/kg) and xylazine (10 mg/kg) and euthanized by transcardial perfusion with 10 ml of cold PBS followed by 10 ml of cold 4% paraformaldehyde in PBS. Brains were extracted and left in a 4% PFA solution in PBS overnight. Fifty µm coronal sections were cut on a vibratome (VT1000, Leica). A subset of tissue was labeled using the following simplified staining protocol. First, brain sections were incubated for 48 hr at 4 °C in the primary antibody diluted in PBST (0.3% Triton-X in PBS). Brain sections were then washed three times for 15 min in PBST before and after incubating for 2 hr at room temperature in the secondary antibody diluted in PBST. The antibodies used in this study and their dilutions are: Rb α-substance P (1:1000 dilution; 20064, Immunostar), Rb α-TH (1:1000 dilution; AB152, Millipore), Dk α-Rb Alexa Fluor 488 (1:2000 dilution; A-210206, Thermo Fisher Scientific), Dk α-Rb Alexa Fluor 647 (1:2000 dilution; A-31573, Thermo Fisher Scientific). Slices were mounted using Fluoromount with a DAPI counterstain (SouthernBiotech) and imaged on an Olympus BX61 VS120 Virtual Slide Scanner and 10 x objective (Olympus). Brains were harvested 21–30 days or 5–7 days after surgery for anterograde and retrograde tracing experiments, respectively. Brains injected for $Ca^{2+}$ imaging were harvested within a week of the last imaging session.

For anterograde tracing quantification, four to six slices containing each of the brain regions of interest (VP, LH, and VTA) were analyzed per animal. To quantify the relative abundance of OT axons in a given brain region, boundaries for the region were drawn on ImageJ Fiji (National Institutes of Health) with reference to the Paxinos and Franklin Mouse Brain Atlas. Afterwards, the percentage of the 16-bit pixels within the boundary that had intensity above 200 was quantified. For retrograde tracing experiments, cells were counted manually every 4th slice.

## Behavior

Mice were water restricted to reach 85–90% of their initial body weight and given access to water for 5 min a day in order to maintain desired weight. Prior to imaging, mice were habituated to the head fixation device (Neurotar) and treadmill for 3–5 days, 15–30 min per session. The treadmill parts were 3D printed using a LCD printer (X1-N, EPAX) from publicly available designs (*Jackson et al., 2018*). During habituation, mice were provided 10% sucrose from the water spout. Walking and licking behaviors were measured using a quadrature encoder (HEDR-5420-es214, Broadcom) and a capacitance sensor (1129_1, Phidgets), respectively. A video feed of the animal's face was also recorded using a camera (acA1300-30um, Basler) with a 8–50 mm zoom lens (C2308ZM50, Arducam) at 20 Hz with infrared illumination (VQ2121, Lorex Technology).

Odor was delivered to the mouse using a custom-built olfactometer. Compressed medical air was split into 2 gas-mass flow controllers (GFC17, Aalborg). One flow controller directed a constant rate of 1.5 L/min to a hollowed out teflon cylinder. The other flow regulator was connected to a three-way solenoid valve (**LHDB1223418H,** The Lee Co.). Prior to odor delivery, the three-way valve directs clean air at 0.5 L/min to the teflon cylinder. During odor delivery, the three-way valve directs air to an

odor manifold, which consists of an array of two-way solenoid valves (LHDB1242115H, The Lee Co.), each connected to a different odor bottle. Depending on the trial type, the appropriate two-way valve opens, directing 0.5 L/min of air flow through the odor bottle containing a kimwipe blotted with 50 µl of diluted odor. All odors were diluted in mineral oil (M5310, Sigma-Aldrich) to 1.5 mmHg. The kinetics and consistency of odor delivery were characterized for 30 trials of terpinene delivery using a miniature Photoionization Detector (mPID; Aurora Scientific, Inc).

During classical conditioning, animals were exposed to the following odors for 2 s: 3-hexanone, 3-heptanone, 3-octanone, α-terpinene, α-pinene, and (R)-(+)-limonene (all odors were purchased from Sigma with the highest available purity). In days 1–3 of training, each of the six odors and associated outcomes were provided 30 times with 12–18 s of inter-trial interval. Hexanone and terpinene were not associated with any outcome, heptanone and pinene were associated with 2 µl of 10% sucrose, and octanone and limonene were associated with a 70 psi airpuff delivered to their hindquarters. Sucrose or airpuff was delivered 100–300ms after the end of odor delivery. Trials were organized into 30 blocks, each of which consisted of one trial of each of the six odors in randomized order. In days 4–6 of training, the outcome contingencies were switched such that heptanone and limonene were not associated with any outcome, octanone and terpinene were associated with 2 µl of 10% sucrose, and hexanone and pinene were associated with 70 psi airpuff.

In the lick-no-lick paradigm, trials were also structured into 30 blocks, each of which consisted of 1 trial of each of the 6 odors in randomized order. Hexanone and terpinene were not associated with any outcome, heptanone and pinene were paired with 2 µl of 10% sucrose at 50% chance, and octanone and limonene were paired with 2 µl of 10% sucrose at 100% chance. 200ms prior to the onset of three of the odors (terpinene, octanone, and limonene), the lick spout was retracted 30 mm away from the animal's mouth using a linear stepper motor (BE073-1, Befenybay) and driver (A4988, BIQU). The lick spout would return to its original position 100ms prior to the earliest possible time of sucrose delivery.

## DeepLabCut

DeepLabCut2.3.3 with Tensorflow 2.12 was used to track 4 points on the periphery of the eye during two-photon $Ca^{2+}$ imaging. The mini-batch k-means clustering method was used to extract a total of 100 frames (20 frames from 5 animals). These frames were labeled and used to train a Deep Neural Network (DNN) model for 100,000 iterations. After the first training session, 20 outlier frames were picked up from each video and added to the training data for a second training session. The area of the eye at a given time point was estimated as an ellipse. For the lick-no-lick paradigm, we used DeepLabCut to track the tip of the tongue, the corner of the mouth, the upper lip and the lower lip. To record licking in the absence of the lick spout, we trained a linear classifier using logistic regression of the following metrics: (1) the confidence score for the tip of the tongue, (2) the confidence score for the corner of the mouth and (3) the Euclidean distance between the upper and lower lip. Data collected from the capacitive lick sensor was used as ground truth for the classifier.

## Two-photon $Ca^{2+}$ imaging in head-fixed, behaving mice

Mice were habituated to the head-fixation setup for 3 days beginning 8–10 weeks after surgery. $Ca^{2+}$ imaging data was acquired using an Olympus FV-MPE-RS Multiphoton microscope with Spectra Physics MaiTai HPDS laser, tuned to 920 nm with 100 fs pulse width at 80 MHz. Each 128x128 pixel scan was acquired with a 20 x air objective (LCPLN20XIR, Olympus), using a Galvo-Galvo scanner at 5 Hz. Stimulus delivery and behavioral measurements were controlled through a custom software written in LabVIEW (National Instruments) and operated through a DAQ (USB-6008, National Instruments). Each imaging session lasted between 30 and 45 min and was synchronized with the stimulus delivery software through a TTL pulse. The imaging depth was manually adjusted to closely match that of the first imaging day such that we recorded from overlapping populations across days of imaging. Animals were excluded from analysis if (a) histology showed that either the GRIN lens or the jGCaMP7s virus was mistargeted or (b) the motion during imaging was too severe for successful motion-correction. Two animals were excluded due to mistargeting and two animals were excluded due to excessive motion.

## Image processing

$Ca^{2+}$ imaging data were first motion-corrected using the non-rigid motion correction algorithm NoRM-Corre (*Pnevmatikakis and Giovannucci, 2017*). Afterwards, neural traces were extracted from the motion-corrected data using constrained nonnegative matrix factorization (CNMF) (*Giovannucci et al., 2019*; *Pnevmatikakis et al., 2016*). Briefly, this algorithm estimates a spatial matrix (analogous to the idea of ROIs in manual processing methods) and a temporal matrix whose products equal the motion-corrected spatiotemporal fluorescence data. Spatial components identified by CNMF were inspected by eye to ensure they were not artifacts. A Gaussian Mixture Model (GMM) was used to estimate the baseline fluorescence of each neuron. To account for potential low-frequency drift in the baseline, the GMM was applied along a moving window of 2500 frames (500 s). The fluorescence of each neuron at each time point $t$ was then normalized to the moving baseline to calculate $\Delta F/F = F_t - F_{baseline}/F_{baseline}$. For analysis comparing the activity of the same neuron across multiple, spatial components from two different imaging days were matched manually. All subsequent analyses were performed using custom code written in MATLAB (R2022b).

## Hierarchical clustering of pooled averaged responses

$\Delta F/F$ in response to all 6 odors on day 6 were averaged across trials then Z-scored. The resulting trial-average values from the following timebins were averaged across time: (1) the first second during each odor, (2) the last second during each odor, and (3) the first second after each odor. The resulting 18-element vectors were sorted into 6 clusters after agglomerative hierarchical clustering using euclidean distance and ward linkage.

## Responsiveness criteria

To determine how many neurons were responsive to a given odor, we compared $\Delta F/F$ at each frame during the 2 s odor period against a pooled distribution of $\Delta F/F$ values from the 2 s prior to odor onset using a Wilcoxon rank sum test. The resulting p-values were evaluated with Holm-Bonferroni correction to ensure that familywise error rate (FWER) was below 0.05. We then calculated the percentage of responsive neurons for each animal to show the mean and the standard error as a function of time. We also counted the number of neurons that were significantly responsive for at least four frames during the odor period to report the total percentage of responsive neurons during odor.

## Single neuron logistic classifiers

To test how reliably a single neuron's fluorescence could discriminate between two odors, we assessed the performance of binary logistic classifiers trained on a single neuron's responses to two odors. For each neuron and odor pair, we averaged the $\Delta F/F$ during the last second of the odor exposure for each trial then Z-scored across all trials. The resulting 60-element vector was used to train a linear classifier using logistic regression. The receiver operator characteristic (ROC) was evaluated for each single neuron pairwise classifier and the area under the curve (AUC) reported. To test if a given pairwise classifier performed significantly better than chance, we compared the accuracy of each classifier against a distribution of 10,000 classifiers trained on shuffled labels.

## Normalized $\Delta\Delta F/F$ correlations

To compare the average response of a neuron to each odor, the trial-averaged $\Delta F/F$ during the last second of odor exposure from each trial was averaged and then subtracted from the trial-averaged $\Delta F/F$ during the 2 s prior to odor delivery. This $\Delta\Delta F/F$ value was scaled to the largest positive $\Delta\Delta F/F$ value of each neuron for all odors. To assess the similarity of the average response to a given pair of odors $i$ and $j$, we looked at the null linear model in which all neurons respond identically to both odors, i.e. $\Delta\Delta F_j/F = \Delta\Delta F_i/F$. To assess how well this describes the data, we report the $R^2$ value of the fit.

## Pairwise euclidean distance

To quantify the differences among population-level responses to the six odors, we quantified the pairwise Euclidean distance between the trajectories of odor responses. First, we subtracted the $\Delta F/F$ values during the 2 s prior to odor delivery from each frame then averaged these values across trials for each odor. The pairwise Euclidean distance at each frame was computed for each odor pair and normalized to the maximum pairwise distance measured in all odor pairs at any time bin. These

calculations were carried out separately for each animal and then averaged across biological replicates to report the mean and the standard error.

## Population pairwise classifiers

To assess the discriminability of odor responses in high-dimensional space, we measured the accuracy of binary classifiers for a given odor pair. At each time point relative to odor delivery, we pooled ΔF/F values from all trials during which either odor was presented. These values were then normalized and used to train a linear classifier using either a logistic regression or a Support Vector Machine (SVM). The accuracy of the classifier was evaluated via 5-fold cross-validation. To test if a given pairwise decoder performed significantly better than chance, we compared the accuracy of each classifier against a distribution of 10,000 classifiers trained on shuffled labels. All classifiers were trained on populations of neurons simultaneously recorded from individual mice. The resulting cross-validated accuracies were averaged across biological replicates to report the mean and the standard error.

## Dimensionality analysis

To quantify the dimensionality of each simultaneously recorded neural population, we calculated its participation ratio (PR). First, we performed principal component analysis of the whole dataset using the singular value decomposition algorithm. The PR was calculated as the square of the sum of the eigenvalues of the covariance matrix divided by the sum of the square of its eigenvalues (*Litwin-Kumar et al., 2017*; *Recanatesi et al., 2019*). To account for the differences in number of recorded neurons across individuals, we bootstrapped the PR by randomly sampling $n$ neurons from each dataset 1000 times and reported the average PR value.

## Statistical analysis

For simple pairwise comparisons, we used Student's t-tests or, when appropriate, Wilcoxon rank sum tests with Benjamini Hochberg correction to adjust for false discovery rate (FDR). For post hoc comparisons following ANOVA's, we used Tukey's honestly significant difference test which adjusts for family-wise error rate (FWER). For linear mixed-effects models with individual animals as random effect, we used the MATLAB fitlme function with maximum likelihood estimation algorithm and Quasi-Newton optimization.

# Acknowledgements

We thank members of Root lab for discussions, M Aoi for discussions on data analysis, and T Komiyama and for comments on the manuscript. This research was supported by grants from the NIH (R00DC014516, R01DC018313), and CMR was a Hellman Fellow.

## Additional information

### Funding

| Funder | Grant reference number | Author |
| --- | --- | --- |
| National Institute on Deafness and Other Communication Disorders | R01DC018313 | Cory M Root |
| National Institute on Deafness and Other Communication Disorders | R00DC014516 | Cory M Root |

The funders had no role in study design, data collection and interpretation, or the decision to submit the work for publication.

### Author contributions

Donghyung Lee, Conceptualization, Resources, Data curation, Software, Formal analysis, Investigation, Methodology, Writing - original draft, Writing - review and editing; Nathan Lau, Lillian Liu, Formal

analysis, Investigation, Methodology; Cory M Root, Conceptualization, Supervision, Funding acquisition, Writing - original draft, Project administration, Writing - review and editing

## Author ORCIDs
Cory M Root  https://orcid.org/0000-0003-0193-8183

Reviewer #1 (Public Review): https://doi.org/10.7554/eLife.90976.4.sa1
Reviewer #2 (Public Review): https://doi.org/10.7554/eLife.90976.4.sa2
Reviewer #3 (Public Review): https://doi.org/10.7554/eLife.90976.4.sa3
Author response https://doi.org/10.7554/eLife.90976.4.sa4

# Additional files

## Supplementary files
• MDAR checklist

## Data availability
Source data for *Figures 1–6* will be published on Dryad at https://doi.org/10.5061/dryad.2547d7x15.

The following dataset was generated:

| Author(s) | Year | Dataset title | Dataset URL | Database and Identifier |
|---|---|---|---|---|
| Lee D, Lau N, Liu L, Root CM | 2024 | Transformation of valence signaling in a striatopallidal circuit | https://doi.org/10.5061/dryad.2547d7x15 | Dryad Digital Repository, 10.5061/dryad.2547d7x15 |

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

# Appendix 1

**Appendix 1—table 1.** Pairwise comparisons of anterograde labeling from OT and AcbSh (*Figure 1C*).

| Group A | Group B | Lower Limit | A-B | Upper Limit | FDR adjusted p-value |
|---|---|---|---|---|---|
| $OT_{D1} \rightarrow VP$ | $Acb_{D1} \rightarrow VP$ | −20.506 | −8.798 | 2.910 | 0.577 |
| $OT_{D1} \rightarrow VP$ | $OT_{D2} \rightarrow VP$ | −0.851 | 9.922 | 20.694 | 0.577 |
| $OT_{D1} \rightarrow VP$ | $Acb_{D2} \rightarrow VP$ | 9.338 | 19.340 | 29.341 | 0.291 |
| $Acb_{D1} \rightarrow VP$ | $OT_{D2} \rightarrow VP$ | 11.163 | 18.719 | 26.275 | 0.161 |
| $Acb_{D1} \rightarrow VP$ | $Acb_{D2} \rightarrow VP$ | 21.729 | 28.137 | 34.545 | 3.394E-02 |
| $OT_{D2} \rightarrow VP$ | $Acb_{D2} \rightarrow VP$ | 4.942 | 9.418 | 13.894 | 0.206 |
| $OT_{D1} \rightarrow VP$ | $Acb_{D1} \rightarrow VP$ | −9.944 | −8.199 | −6.453 | 2.223E-02 |
| $OT_{D1} \rightarrow LH$ | $OT_{D2} \rightarrow LH$ | −0.045 | 0.564 | 1.173 | 0.577 |
| $OT_{D1} \rightarrow LH$ | $Acb_{D2} \rightarrow LH$ | −0.016 | 0.591 | 1.198 | 0.577 |
| $Acb_{D1} \rightarrow LH$ | $OT_{D2} \rightarrow LH$ | 7.125 | 8.763 | 10.401 | 2.223E-02 |
| $Acb_{D1} \rightarrow LH$ | $Acb_{D2} \rightarrow LH$ | 7.153 | 8.790 | 10.426 | 2.223E-02 |
| $OT_{D2} \rightarrow LH$ | $Acb_{D2} \rightarrow LH$ | −0.029 | 0.027 | 0.082 | 0.655 |
| $OT_{D1} \rightarrow VTA$ | $Acb_{D1} \rightarrow VTA$ | −17.357 | −14.504 | −11.650 | 2.223E-02 |
| $OT_{D1} \rightarrow VTA$ | $OT_{D2} \rightarrow VTA$ | −0.022 | 0.183 | 0.387 | 0.577 |

**Appendix 1—table 2.** Pairwise comparisons of retrograde labeling from vlVP and dmVP (*Figure 1F*).

| Group A | Group B | Lower Limit | A-B | Upper Limit | FDR adjusted p-value |
|---|---|---|---|---|---|
| AI→vlVP | AI→dmVP | 78.249 | 148.500 | 218.751 | 0.116 |
| Acb→vlVP | Acb→dmVP | −185.929 | −123.250 | −60.571 | 0.116 |
| LS→vlVP | LS→dmVP | −130.496 | −118.750 | −107.004 | 3.27E-04 |
| OFC→vlVP | OFC→dmVP | −167.021 | −132.000 | −96.979 | 1.86E-02 |
| OT→vlVP | OT→dmVP | 179.793 | 221.750 | 263.707 | 5.57E-03 |
| Pir→vlVP | Pir→dmVP | −71.946 | −21.500 | 28.946 | 0.685 |

**Appendix 1—table 3.** Pairwise comparisons of retrograde labeling from VTA (*Figure 1I*).

| Group A | Group B | Lower Limit | A-B | Upper Limit | FDR adjusted p-value |
|---|---|---|---|---|---|
| OT→VTA | AcbSh→VTA | −735.101 | −575 | −414.899 | 3.44E-02 |
| OT→VTA | AcbC→VTA | −1027.381 | −915 | −802.619 | 3.71E-03 |
| AcbC→VTA | AcbSh→VTA | 144.452 | 340 | 535.548 | 0.157 |

**Appendix 1—table 4.** Two-way ANOVA for effect of day or lens placement on licking accuracy (*Figure 2H*).

| Source | Sum Sq. | d.f. | Mean Sq. | F | Prob >F |
|---|---|---|---|---|---|
| day | 4.301 | 5 | 0.860 | 27.638 | 2.29E-16 |
| region | 0.143 | 2 | 0.072 | 2.301 | 0.106 |
| day:region | 0.251 | 10 | 0.025 | 0.806 | 0.623 |
| Error | 2.583 | 83 | 0.031 | | |
| Total | 7.305 | 100 | | | |

**Appendix 1—table 5.** Post hoc pairwise comparisons of licking accuracy across imaging days (*Figure 2H*).

| Group A | Group B | Lower Limit | A-B | Upper Limit | p-value |
|---------|---------|-------------|-----|-------------|---------|
| day 1 | day 2 | –0.385 | –0.208 | –0.031 | 1.18E-02 |
| day 1 | day 3 | –0.632 | –0.452 | –0.272 | 2.03E-09 |
| day 1 | day 4 | –0.134 | 0.043 | 0.220 | 0.980 |
| day 1 | day 5 | –0.558 | –0.381 | –0.203 | 2.30E-07 |
| day 1 | day 6 | –0.650 | –0.473 | –0.295 | 2.58E-10 |
| day 2 | day 3 | –0.424 | –0.244 | –0.064 | 2.18E-03 |
| day 2 | day 4 | 0.074 | 0.251 | 0.429 | 1.14E-03 |
| day 2 | day 5 | –0.350 | –0.172 | 0.005 | 0.061 |
| day 2 | day 6 | –0.441 | –0.264 | –0.087 | 5.33E-04 |
| day 3 | day 4 | 0.315 | 0.495 | 0.675 | 8.31E-11 |
| day 3 | day 5 | –0.109 | 0.071 | 0.251 | 0.856 |
| day 3 | day 6 | –0.200 | –0.021 | 0.159 | 0.999 |
| day 4 | day 5 | –0.601 | –0.424 | –0.247 | 1.00E-08 |
| day 4 | day 6 | –0.693 | –0.516 | –0.338 | 9.77E-12 |
| day 5 | day 6 | –0.269 | –0.092 | 0.085 | 0.657 |

**Appendix 1—table 6.** 2-way ANOVA for effect of day or lens placement on percentage of neurons responsive to a single odor (*Figure 3E*).

| Source | Sum Sq. | d.f. | Mean Sq. | F | Prob >F |
|--------|---------|------|----------|---|---------|
| day | 0.05917 | 2 | 0.02958 | 2.99563 | 0.06079 |
| region | 0.03696 | 2 | 0.01848 | 1.87142 | 0.16651 |
| day:region | 0.06481 | 4 | 0.01620 | 1.64062 | 0.18197 |
| Error | 0.41479 | 42 | 0.00988 | | |
| Total | 0.57170 | 50 | | | |

**Appendix 1—table 7.** Post hoc pairwise comparisons of percentage of neurons responsive to a single odor across imaging days and region (*Figure 3E*).

| Group A | Group B | Lower Limit | A-B | Upper Limit | p-value |
|---------|---------|-------------|-----|-------------|---------|
| d1,D2 OT | d3,D2 OT | –0.140 | 0.048 | 0.236 | 0.995 |
| d1,D2 OT | d6,D2 OT | –0.164 | 0.024 | 0.211 | 1.000 |
| d3,D2 OT | d6,D2 OT | –0.212 | –0.024 | 0.163 | 1.000 |
| d1,D1 OT | d3,D1 OT | –0.287 | –0.099 | 0.088 | 0.724 |
| d1,D1 OT | d6,D1 OT | –0.328 | –0.141 | 0.047 | 0.285 |
| d3,D1 OT | d6,D1 OT | –0.229 | –0.041 | 0.146 | 0.998 |
| d1,VP | d3,VP | –0.321 | –0.115 | 0.090 | 0.662 |
| d1,VP | d6,VP | –0.335 | –0.129 | 0.076 | 0.513 |
| d3,VP | d6,VP | –0.220 | –0.014 | 0.191 | 1.000 |
| d1,D2 OT | d1,D1 OT | –0.115 | 0.072 | 0.260 | 0.937 |
| d1,D2 OT | d1,VP | –0.056 | 0.141 | 0.338 | 0.341 |
| d1,D1 OT | d1,VP | –0.128 | 0.069 | 0.265 | 0.964 |

*Appendix 1—table 7 Continued on next page*

*Appendix 1—table 7 Continued*

| Group A | Group B | Lower Limit | A-B | Upper Limit | p-value |
|---|---|---|---|---|---|
| d3,D2 OT | d3,D1 OT | –0.263 | –0.075 | 0.112 | 0.923 |
| d3,D2 OT | d3,VP | –0.219 | –0.022 | 0.175 | 1.000 |
| d3,D1 OT | d3,VP | –0.144 | 0.053 | 0.250 | 0.993 |
| d6,D2 OT | d6,D1 OT | –0.280 | –0.092 | 0.095 | 0.796 |
| d6,D2 OT | d6,VP | –0.209 | –0.012 | 0.184 | 1.000 |
| d6,D1 OT | d6,VP | –0.117 | 0.080 | 0.276 | 0.918 |

**Appendix 1—table 8.** Two-way ANOVA for effect of day or lens placement on percentage of neurons responsive to three or more odors (*Figure 3E*).

| Source | Sum Sq. | d.f. | Mean Sq. | F | Prob >F |
|---|---|---|---|---|---|
| day | 0.10897 | 2 | 0.05448 | 4.58607 | 0.01580 |
| region | 0.00469 | 2 | 0.00234 | 0.19732 | 0.82168 |
| day:region | 0.06640 | 4 | 0.01660 | 1.39730 | 0.25144 |
| Error | 0.49897 | 42 | 0.01188 | | |
| Total | 0.66598 | 50 | | | |

**Appendix 1—table 9.** Post hoc pairwise comparisons of percentage of neurons responsive to three or more odors across imaging days and region (*Figure 3E*).

| Group A | Group B | Lower Limit | A-B | Upper Limit | p-value |
|---|---|---|---|---|---|
| d1,D2 OT | d3,D2 OT | –0.237 | –0.031 | 0.175 | 1.000 |
| d1,D2 OT | d6,D2 OT | –0.246 | –0.040 | 0.165 | 0.999 |
| d3,D2 OT | d6,D2 OT | –0.215 | –0.009 | 0.196 | 1.000 |
| d1,D1 OT | d3,D1 OT | –0.230 | –0.024 | 0.182 | 1.000 |
| d1,D1 OT | d6,D1 OT | –0.269 | –0.063 | 0.143 | 0.984 |
| d3,D1 OT | d6,D1 OT | –0.245 | –0.039 | 0.167 | 0.999 |
| d1,VP | d3,VP | –0.406 | –0.181 | 0.044 | 0.207 |
| d1,VP | d6,VP | –0.453 | –0.228 | –0.003 | 0.046 |
| d3,VP | d6,VP | –0.272 | –0.047 | 0.178 | 0.999 |
| d1,D2 OT | d1,D1 OT | –0.220 | –0.014 | 0.191 | 1.000 |
| d1,D2 OT | d1,VP | –0.124 | 0.092 | 0.308 | 0.894 |
| d1,D1 OT | d1,VP | –0.109 | 0.106 | 0.322 | 0.793 |
| d3,D2 OT | d3,D1 OT | –0.213 | –0.007 | 0.198 | 1.000 |
| d3,D2 OT | d3,VP | –0.274 | –0.058 | 0.158 | 0.993 |
| d3,D1 OT | d3,VP | –0.266 | –0.051 | 0.165 | 0.997 |
| d6,D2 OT | d6,D1 OT | –0.243 | –0.037 | 0.168 | 1.000 |
| d6,D2 OT | d6,VP | –0.311 | –0.096 | 0.120 | 0.871 |
| d6,D1 OT | d6,VP | –0.274 | –0.059 | 0.157 | 0.993 |

*Appendix 1—table 10 Continued on next page*

**Appendix 1—table 10.** Two-way ANOVA for effect of day or lens placement on percentage of neurons responsive to both S-cues (*Figure 3E*).

| Source | Sum Sq. | d.f. | Mean Sq. | F | Prob >F |
|---|---|---|---|---|---|
| day | 0.11460 | 2 | 0.05730 | 6.60475 | 0.00321 |
| region | 0.18370 | 2 | 0.09185 | 10.58706 | 0.00019 |
| day:region | 0.16522 | 4 | 0.04131 | 4.76096 | 0.00294 |
| Error | 0.36438 | 42 | 0.00868 | | |
| Total | 0.80767 | 50 | | | |

**Appendix 1—table 11.** Post hoc pairwise comparisons of percentage of neurons responsive to both S-cues across imaging days and region (*Figure 3E*).

| Group A | Group B | Lower Limit | A-B | Upper Limit | p-value |
|---|---|---|---|---|---|
| d1,D2 OT | d3,D2 OT | –0.139 | 0.037 | 0.213 | 0.999 |
| d1,D2 OT | d6,D2 OT | –0.200 | –0.024 | 0.151 | 1.000 |
| d3,D2 OT | d6,D2 OT | –0.237 | –0.062 | 0.114 | 0.963 |
| d1,D1 OT | d3,D1 OT | –0.239 | –0.063 | 0.112 | 0.957 |
| d1,D1 OT | d6,D1 OT | –0.196 | –0.020 | 0.156 | 1.000 |
| d3,D1 OT | d6,D1 OT | –0.133 | 0.043 | 0.219 | 0.996 |
| d1,VP | d3,VP | –0.441 | –0.248 | –0.056 | 3.79E-03 |
| d1,VP | d6,VP | –0.473 | –0.281 | –0.088 | 7.12E-04 |
| d3,VP | d6,VP | –0.225 | –0.032 | 0.160 | 1.000 |
| d1,D2 OT | d1,D1 OT | –0.189 | –0.013 | 0.163 | 1.000 |
| d1,D2 OT | d1,VP | –0.151 | 0.033 | 0.217 | 1.000 |
| d1,D1 OT | d1,VP | –0.138 | 0.046 | 0.230 | 0.996 |
| d3,D2 OT | d3,D1 OT | –0.289 | –0.114 | 0.062 | 0.480 |
| d3,D2 OT | d3,VP | –0.437 | –0.252 | –0.068 | 1.72E-03 |
| d3,D1 OT | d3,VP | –0.323 | –0.139 | 0.045 | 0.279 |
| d6,D2 OT | d6,D1 OT | –0.185 | –0.009 | 0.167 | 1.000 |
| d6,D2 OT | d6,VP | –0.408 | –0.223 | –0.039 | 7.94E-03 |
| d6,D1 OT | d6,VP | –0.399 | –0.214 | –0.030 | 1.23E-02 |

**Appendix 1—table 12.** Two-way ANOVA for effect of day or lens placement on percentage of neurons with auROC >0.75 for {$S_K$ vs. $P_K$} (*Figure 3I*).

| Source | Sum Sq. | d.f. | Mean Sq. | F | Prob >F |
|---|---|---|---|---|---|
| day | 0.85429 | 2 | 0.42714 | 46.23443 | 2.439E-11 |
| region | 0.32224 | 2 | 0.16112 | 17.43954 | 3.064E-06 |
| day:region | 0.40111 | 4 | 0.10028 | 10.85411 | 3.918E-06 |
| Error | 0.38802 | 42 | 0.00924 | | |
| Total | 1.87808 | 50 | | | |

*Appendix 1—table 13 Continued on next page*

**Appendix 1—table 13.** Post hoc comparisons of percentage of neurons with auROC >0.75 for {$S_K$ vs. $P_K$} across imaging day and region (*Figure 3I*).

| Group A | Group B | Lower Limit | A-B | Upper Limit | p-value |
|---|---|---|---|---|---|
| d1,D2 OT | d3,D2 OT | –0.259 | –0.078 | 0.103 | 0.890 |
| d1,D2 OT | d6,D2 OT | –0.279 | –0.097 | 0.084 | 0.709 |
| d3,D2 OT | d6,D2 OT | –0.201 | –0.020 | 0.162 | 1.000 |
| d1,D1 OT | d3,D1 OT | –0.293 | –0.112 | 0.069 | 0.541 |
| d1,D1 OT | d6,D1 OT | –0.417 | –0.236 | –0.054 | 0.003 |
| d3,D1 OT | d6,D1 OT | –0.305 | –0.124 | 0.058 | 0.407 |
| d1,VP | d3,VP | –0.476 | –0.278 | –0.079 | 0.001 |
| d1,VP | d6,VP | –0.820 | –0.621 | –0.423 | 2.00E-11 |
| d3,VP | d6,VP | –0.542 | –0.344 | –0.145 | 4.09E-05 |
| d1,D2 OT | d1,D1 OT | –0.170 | 0.011 | 0.192 | 1.000 |
| d1,D2 OT | d1,VP | –0.141 | 0.049 | 0.239 | 0.995 |
| d1,D1 OT | d1,VP | –0.152 | 0.038 | 0.228 | 0.999 |
| d3,D2 OT | d3,D1 OT | –0.204 | –0.023 | 0.158 | 1.000 |
| d3,D2 OT | d3,VP | –0.341 | –0.151 | 0.039 | 0.221 |
| d3,D1 OT | d3,VP | –0.318 | –0.128 | 0.062 | 0.427 |
| d6,D2 OT | d6,D1 OT | –0.309 | –0.127 | 0.054 | 0.370 |
| d6,D2 OT | d6,VP | –0.665 | –0.475 | –0.285 | 1.15E-08 |
| d6,D1 OT | d6,VP | –0.538 | –0.348 | –0.158 | 1.43E-05 |

**Appendix 1—table 14.** Two-way ANOVA for effect of day or lens placement on percentage of neurons with auROC >0.75 for {$S_K$ vs. $X_K$} (*Figure 3J*).

| Source | Sum Sq. | d.f. | Mean Sq. | F | Prob >F |
|---|---|---|---|---|---|
| day | 0.551 | 2 | 0.275 | 28.099 | 1.794E-08 |
| region | 0.377 | 2 | 0.188 | 19.265 | 1.156E-06 |
| day:region | 0.364 | 4 | 0.0912 | 9.310 | 1.764E-05 |
| Error | 0.411 | 42 | 0.009 | | |
| Total | 1.637 | 50 | | | |

**Appendix 1—table 15.** Post hoc comparisons of percentage of neurons with auROC >0.75 for {$S_K$ vs. $X_K$} across imaging day and region (*Figure 3J*).

| Group A | Group B | Lower Limit | A-B | Upper Limit | p-value |
|---|---|---|---|---|---|
| d1,D2 OT | d3,D2 OT | –0.189 | –0.002 | 0.184 | 1.000 |
| d1,D2 OT | d6,D2 OT | –0.238 | –0.052 | 0.135 | 0.992 |
| d3,D2 OT | d6,D2 OT | –0.236 | –0.049 | 0.137 | 0.994 |
| d1,D1 OT | d3,D1 OT | –0.313 | –0.126 | 0.061 | 0.421 |
| d1,D1 OT | d6,D1 OT | –0.356 | –0.170 | 0.017 | 0.102 |
| d3,D1 OT | d6,D1 OT | –0.230 | –0.044 | 0.143 | 0.997 |
| d1,VP | d3,VP | –0.454 | –0.250 | –0.045 | 7.23E-03 |
| d1,VP | d6,VP | –0.750 | –0.545 | –0.341 | 2.05E-09 |
| d3,VP | d6,VP | –0.500 | –0.295 | –0.091 | 8.14E-04 |

*Appendix 1—table 15 Continued on next page*

*Appendix 1—table 15 Continued*

| Group A | Group B | Lower Limit | A-B | Upper Limit | p-value |
|---|---|---|---|---|---|
| d1,D2 OT | d1,D1 OT | −0.219 | −0.033 | 0.154 | 1.000 |
| d1,D2 OT | d1,VP | −0.163 | 0.033 | 0.229 | 1.000 |
| d1,D1 OT | d1,VP | −0.130 | 0.066 | 0.261 | 0.972 |
| d3,D2 OT | d3,D1 OT | −0.343 | −0.156 | 0.031 | 0.167 |
| d3,D2 OT | d3,VP | −0.410 | −0.214 | −0.019 | 2.26E-02 |
| d3,D1 OT | d3,VP | −0.254 | −0.058 | 0.138 | 0.987 |
| d6,D2 OT | d6,D1 OT | −0.337 | −0.151 | 0.036 | 0.204 |
| d6,D2 OT | d6,VP | −0.656 | −0.461 | −0.265 | 5.39E-08 |
| d6,D1 OT | d6,VP | −0.506 | −0.310 | −0.114 | 1.95E-04 |

**Appendix 1—table 16.** Two-way ANOVA for effect of day or lens placement on percentage of neurons with auROC >0.75 for {$S_K$ vs. $S_T$} (***Figure 3K***).

| Source | Sum Sq. | d.f. | Mean Sq. | F | Prob >F |
|---|---|---|---|---|---|
| day | 0.01400 | 2 | 0.00700 | 0.45251 | 0.63909 |
| region | 0.08031 | 2 | 0.04015 | 2.59595 | 0.08650 |
| day:region | 0.01282 | 4 | 0.00321 | 0.20726 | 0.93297 |
| Error | 0.64967 | 42 | 0.01547 | | |
| Total | 0.75527 | 50 | | | |

**Appendix 1—table 17.** Post hoc comparisons of percentage of neurons with auROC >0.75 for {$S_K$ vs. $S_T$} across imaging day and region (***Figure 3K***).

| Group A | Group B | Lower Limit | A-B | Upper Limit | p-value |
|---|---|---|---|---|---|
| d1,D2 OT | d3,D2 OT | −0.237 | −0.002 | 0.232 | 1.000 |
| d1,D2 OT | d6,D2 OT | −0.276 | −0.041 | 0.193 | 1.000 |
| d3,D2 OT | d6,D2 OT | −0.274 | −0.039 | 0.196 | 1.000 |
| d1,D1 OT | d3,D1 OT | −0.255 | −0.020 | 0.214 | 1.000 |
| d1,D1 OT | d6,D1 OT | −0.233 | 0.001 | 0.236 | 1.000 |
| d3,D1 OT | d6,D1 OT | −0.213 | 0.022 | 0.257 | 1.000 |
| d1,VP | d3,VP | −0.316 | −0.059 | 0.198 | 0.998 |
| d1,VP | d6,VP | −0.336 | −0.079 | 0.178 | 0.983 |
| d3,VP | d6,VP | −0.277 | −0.020 | 0.237 | 1.000 |
| d1,D2 OT | d1,D1 OT | −0.267 | −0.032 | 0.202 | 1.000 |
| d1,D2 OT | d1,VP | −0.142 | 0.104 | 0.350 | 0.900 |
| d1,D1 OT | d1,VP | −0.110 | 0.136 | 0.382 | 0.678 |
| d3,D2 OT | d3,D1 OT | −0.285 | −0.050 | 0.184 | 0.999 |
| d3,D2 OT | d3,VP | −0.199 | 0.047 | 0.293 | 0.999 |
| d3,D1 OT | d3,VP | −0.149 | 0.097 | 0.344 | 0.928 |
| d6,D2 OT | d6,D1 OT | −0.224 | 0.011 | 0.245 | 1.000 |
| d6,D2 OT | d6,VP | −0.180 | 0.066 | 0.312 | 0.993 |
| d6,D1 OT | d6,VP | −0.191 | 0.055 | 0.302 | 0.998 |

**Appendix 1—table 18.** Pairwise comparisons of $|\Delta\Delta F_{day3}|-|\Delta\Delta F_{day1}|$ across regions (*Figure 4I*).

| Group A | Group B | Lower Limit | A-B | Upper Limit | FDR adjusted p-value |
|---|---|---|---|---|---|
| D2 OT | D1 OT | –0.204 | –0.138 | –0.073 | 4.37E-02 |
| D2 OT | VP | –0.275 | –0.214 | –0.153 | 1.08E-03 |
| D1 OT | VP | –0.120 | –0.075 | –0.031 | 0.141 |

**Appendix 1—table 19.** One sample t-tests of $|\Delta\Delta F_{day3}|-|\Delta\Delta F_{day1}|$ in different regions (*Figure 4I*).

| Population | Lower Limit | Mean | Upper Limit | FDR adjusted p-value |
|---|---|---|---|---|
| D2 OT | –0.146 | –6.51E-02 | 1.58E-02 | 0.259 |
| D1 OT | 2.78E-02 | 7.83E-02 | 0.129 | 4.48E-02 |
| VP | 0.133 | 0.174 | 0.215 | 2.49E-07 |

**Appendix 1—table 20.** One-way ANOVA for effect of region on {S vs. X|P} linear classifier accuracy (*Figure 5G*).

| Source | Sum Sq. | d.f. | Mean Sq. | F | Prob >F |
|---|---|---|---|---|---|
| region | 0.018 | 2 | 9.10E-03 | 9.569 | 2.40E-03 |
| Error | 0.013 | 14 | 9.51E-04 | | |
| Total | 0.032 | 16 | | | |

**Appendix 1—table 21.** Post hoc comparisons of {S vs. X|P} linear classifier accuracy across regions (*Figure 5G*).

| Group A | Group B | Lower Limit | A-B | Upper Limit | p-value |
|---|---|---|---|---|---|
| D2 OT | D1 OT | –0.114 | –0.067 | –0.020 | 5.57E-03 |
| D2 OT | VP | –0.119 | –0.070 | –0.021 | 5.65E-03 |
| D1 OT | VP | –0.052 | –0.003 | 0.046 | 0.985 |

**Appendix 1—table 22.** One-way ANOVA for effect of region on generalized {S vs. X|P} linear classifier accuracy (*Figure 5G*).

| Source | Sum Sq. | d.f. | Mean Sq. | F | Prob >F |
|---|---|---|---|---|---|
| region | 0.134 | 2 | 0.067 | 14.136 | 4.37E-04 |
| Error | 0.066 | 14 | 0.005 | | |
| Total | 0.201 | 16 | | | |

**Appendix 1—table 23.** Post hoc comparisons of generalized {S vs. X|P} linear classifier accuracy across regions (*Figure 5G*).

| Group A | Group B | Lower Limit | A-B | Upper Limit | p-value |
|---|---|---|---|---|---|
| D2 OT | D1 OT | –0.153 | –0.049 | 0.055 | 0.451 |
| D2 OT | VP | –0.323 | –0.214 | –0.105 | 4.17E-04 |
| D1 OT | VP | –0.274 | –0.165 | –0.056 | 3.84E-03 |

**Appendix 1—table 24.** Two-way ANOVA for effect of imaging days or region on normalized PR (*Figure 5I*).

| Source | Sum Sq. | d.f. | Mean Sq. | F | Prob >F |
|---|---|---|---|---|---|
| days | 0.775 | 2 | 0.387 | 0.277 | 0.759 |
| region | 50.226 | 2 | 25.113 | 17.969 | 2.704E-06 |
| days:region | 14.484 | 4 | 3.621 | 2.591 | 0.051 |

**Appendix 1—table 25.** Post hoc comparisons of normalized PR across imaging day and region (*Figure 5I*).

| Group A | Group B | Lower Limit | A-B | Upper Limit | p-value |
|---|---|---|---|---|---|
| d1,D2 OT | d3,D2 OT | –2.938 | –0.592 | 1.754 | 0.995 |
| d1,D2 OT | d6,D2 OT | –3.700 | –1.355 | 0.991 | 0.623 |
| d3,D2 OT | d6,D2 OT | –2.999 | –0.762 | 1.474 | 0.968 |
| d1,D1 OT | d3,D1 OT | –1.948 | 0.289 | 2.526 | 1 |
| d1,D1 OT | d6,D1 OT | –1.965 | 0.272 | 2.509 | 1 |
| d3,D1 OT | d6,D1 OT | –2.254 | –0.017 | 2.220 | 1 |
| d1,VP | d3,VP | –2.404 | 0.195 | 2.794 | 1 |
| d1,VP | d6,VP | –0.783 | 1.816 | 4.415 | 0.372 |
| d3,VP | d6,VP | –0.829 | 1.621 | 4.071 | 0.445 |
| d1,D2 OT | d3,D1 OT | –3.316 | –0.971 | 1.375 | 0.907 |
| d1,D2 OT | d1,VP | –1.921 | 0.678 | 3.277 | 0.994 |
| d1,D1 OT | d1,VP | –0.563 | 1.938 | 4.438 | 0.245 |
| d3,D2 OT | d3,D1 OT | –2.615 | –0.378 | 1.858 | 1 |
| d3,D2 OT | d3,VP | –0.881 | 1.465 | 3.811 | 0.522 |
| d3,D1 OT | d3,VP | –0.502 | 1.843 | 4.189 | 0.229 |
| d6,D2 OT | d6,D1 OT | –1.870 | 0.367 | 2.604 | 1 |
| d6,D2 OT | d6,VP | 1.503 | 3.848 | 6.194 | 1.14E-04 |
| d6,D1 OT | d6,VP | 1.135 | 3.481 | 5.827 | 5.67E-04 |

**Appendix 1—table 26.** One-way ANOVA for effect of region on $\{S_K$ vs. $P_K\}$ linear classifier accuracy trained on PC1 (*Figure 5L*).

| Source | Sum Sq. | d.f. | Mean Sq. | F | Prob >F |
|---|---|---|---|---|---|
| region | 0.209 | 2 | 0.104 | 20.965 | 6.16E-05 |
| Error | 0.070 | 14 | 0.005 | | |
| Total | 0.279 | 16 | | | |

**Appendix 1—table 27.** Post hoc comparisons of $\{S_K$ vs. $P_K\}$ linear classifier accuracy trained on PC1 across regions (*Figure 5L*).

| Group A | Group B | Lower Limit | A-B | Upper Limit | p-value |
|---|---|---|---|---|---|
| D2 OT | D1 OT | –0.172 | –0.066 | 0.041 | 0.273 |
| D2 OT | VP | –0.380 | –0.268 | –0.157 | 5.64E-05 |
| D1 OT | VP | –0.315 | –0.203 | –0.091 | 8.57E-04 |

**Appendix 1—table 28.** One-way ANOVA for effect of region on $\{S_K$ vs. $S_T\}$ linear classifier accuracy trained on PC1-PC15 (*Figure 5L*).

| Source | Sum Sq. | d.f. | Mean Sq. | F | Prob >F |
|---|---|---|---|---|---|
| region | 0.019 | 2 | 0.009 | 0.646 | 0.539 |
| Error | 0.206 | 14 | 0.015 | | |
| Total | 0.225 | 16 | | | |

**Appendix 1—table 29.** Post hoc comparisons of $\{S_K$ vs. $S_T\}$ linear classifier accuracy trained on PC1-PC15 across regions (**Figure 5L**).

| Group A | Group B | Lower Limit | A-B | Upper Limit | p-value |
|---------|---------|-------------|-----|-------------|---------|
| D2 OT | D1 OT | –0.203 | –0.019 | 0.164 | 0.958 |
| D2 OT | VP | –0.131 | 0.061 | 0.253 | 0.687 |
| D1 OT | VP | –0.111 | 0.081 | 0.273 | 0.529 |

**Appendix 1—table 30.** Two-way ANOVA for effect of lick spout presence and sucrose contingency on anticipatory licking (**Figure 6C**).

| Source | Sum Sq. | d.f. | Mean Sq. | F | Prob >F |
|--------|---------|------|----------|---|---------|
| spout | 17.176 | 1 | 17.176 | 50.125 | 2.56E-07 |
| S% | 19.415 | 2 | 9.707 | 28.329 | 4.82E-07 |
| spout:S% | 11.533 | 2 | 5.767 | 16.829 | 2.71E-05 |
| Error | 8.224 | 24 | 0.343 | | |
| Total | 56.348 | 29 | | | |

**Appendix 1—table 31.** Post hoc comparisons of anticipatory licking across spout presence and sucrose contingency (**Figure 6C**).

| Group A | Group B | Lower Limit | A-B | Upper Limit | p-value |
|---------|---------|-------------|-----|-------------|---------|
| spout = 0, S0.0 | spout = 1, S0.0 | –1.205 | –0.060 | 1.085 | 1 |
| spout = 0, S0.0 | spout = 0, S0.5 | –1.335 | –0.190 | 0.955 | 0.995 |
| spout = 0, S0.0 | spout = 1, S0.5 | –2.725 | –1.580 | –0.435 | 3.24E-03 |
| spout = 0, S0.0 | spout = 0, S1.0 | –1.595 | –0.450 | 0.695 | 0.825 |
| spout = 0, S0.0 | spout = 1, S1.0 | –4.685 | –3.540 | –2.395 | 1.66E-08 |
| spout = 1, S0.0 | spout = 0, S0.5 | –1.275 | –0.130 | 1.015 | 0.999 |
| spout = 1, S0.0 | spout = 1, S0.5 | –2.665 | –1.520 | –0.375 | 4.80E-03 |
| spout = 1, S0.0 | spout = 0, S1.0 | –1.535 | –0.390 | 0.755 | 0.895 |
| spout = 1, S0.0 | spout = 1, S1.0 | –4.625 | –3.480 | –2.335 | 2.29E-08 |
| spout = 0, S0.5 | spout = 1, S0.5 | –2.535 | –1.390 | –0.245 | 1.11E-02 |
| spout = 0, S0.5 | spout = 0, S1.0 | –1.405 | –0.260 | 0.885 | 0.980 |
| spout = 0, S0.5 | spout = 1, S1.0 | –4.495 | –3.350 | –2.205 | 4.70E-08 |
| spout = 1, S0.5 | spout = 0, S1.0 | –0.015 | 1.130 | 2.275 | 5.44E-02 |
| spout = 1, S0.5 | spout = 1, S1.0 | –3.105 | –1.960 | –0.815 | 2.58E-04 |
| spout = 0, S1.0 | spout = 1, S1.0 | –4.235 | –3.090 | –1.945 | 2.07E-07 |

**Appendix 1—table 32.** Two-way ANOVA for effect of imaging day and valence of odor on velocity during cue presentation (**Figure 2—figure supplement 5E**).

| Source | Sum Sq. | d.f. | Mean Sq. | F | Prob >F |
|--------|---------|------|----------|---|---------|
| days | 466.659 | 5 | 93.332 | 1.199 | 0.308 |
| valence | 217.697 | 2 | 108.849 | 1.399 | 0.248 |
| days:valence | 671.044 | 10 | 67.104 | 0.862 | 0.569 |
| Error | 35948.767 | 462 | 77.811 | | |
| Total | 37302.764 | 479 | | | |

**Appendix 1—table 33.** Two-way ANOVA for effect of imaging day and valence of odor on velocity during unconditioned stimulus (*Figure 2—figure supplement 5E*).

| Source | Sum Sq. | d.f. | Mean Sq. | F | Prob >F |
|---|---|---|---|---|---|
| days | 851.615 | 5 | 170.323 | 0.660 | 0.654 |
| valence | 29846.379 | 2 | 14923.189 | 57.811 | 3.91E-23 |
| days:valence | 2979.836 | 10 | 297.984 | 1.154 | 0.320 |
| Error | 119259.047 | 462 | 258.136 | | |
| Total | 153052.200 | 479 | | | |

**Appendix 1—table 34.** Post hoc comparisons of velocity during unconditioned stimulus across imaging days and valence of odor (*Figure 2—figure supplement 5E*).

| Group A | Group B | Lower Limit | A-B | Upper Limit | p-value |
|---|---|---|---|---|---|
| d1,P | d1,X | 5.990 | 20.970 | 35.951 | 1.51E-04 |
| d1,P | d1,S | 10.286 | 25.266 | 40.246 | 5.70E-07 |
| d2,P | d2,X | –3.390 | 11.591 | 26.571 | 0.381 |
| d2,P | d2,S | –2.310 | 12.670 | 27.651 | 0.225 |
| d3,P | d3,X | –4.604 | 10.942 | 26.488 | 0.565 |
| d3,P | d3,S | –1.079 | 14.466 | 30.012 | 0.104 |
| d4,P | d4,X | –4.253 | 11.292 | 26.838 | 0.504 |
| d4,P | d4,S | –1.729 | 13.817 | 29.363 | 0.155 |
| d5,P | d5,X | –2.610 | 12.936 | 28.482 | 0.251 |
| d5,P | d5,S | 4.085 | 19.631 | 35.177 | 1.45E-03 |
| d6,P | d6,X | –1.619 | 13.927 | 29.473 | 0.145 |
| d6,P | d6,S | 10.737 | 26.283 | 41.829 | 5.22E-07 |
| d1,P | d2,P | –5.400 | 9.580 | 24.560 | 0.733 |
| d1,P | d3,P | –5.063 | 10.203 | 25.469 | 0.660 |
| d1,P | d4,P | –4.305 | 10.961 | 26.227 | 0.526 |
| d1,P | d5,P | –6.954 | 8.311 | 23.577 | 0.913 |
| d1,P | d6,P | –10.637 | 4.628 | 19.894 | 1 |
| d2,P | d3,P | –14.643 | 0.623 | 15.889 | 1 |
| d2,P | d4,P | –13.885 | 1.381 | 16.647 | 1 |
| d2,P | d5,P | –16.534 | –1.269 | 13.997 | 1 |
| d2,P | d6,P | –20.217 | –4.951 | 10.314 | 1 |
| d3,P | d4,P | –14.788 | 0.758 | 16.304 | 1 |
| d3,P | d5,P | –17.437 | –1.892 | 13.654 | 1 |
| d3,P | d6,P | –21.120 | –5.574 | 9.971 | 0.999 |
| d4,P | d5,P | –18.196 | –2.650 | 12.896 | 1 |
| d4,P | d6,P | –21.878 | –6.333 | 9.213 | 0.995 |
| d5,P | d6,P | –19.229 | –3.683 | 11.863 | 1 |
| d1,X | d2,X | –14.780 | 0.200 | 15.181 | 1 |
| d1,X | d3,X | –15.091 | 0.175 | 15.441 | 1 |
| d1,X | d4,X | –13.982 | 1.283 | 16.549 | 1 |

*Appendix 1—table 34 Continued on next page*

*Appendix 1—table 34 Continued*

| Group A | Group B | Lower Limit | A-B | Upper Limit | p-value |
|---|---|---|---|---|---|
| d1,X | d5,X | −14.989 | 0.277 | 15.543 | 1 |
| d1,X | d6,X | −17.680 | −2.414 | 12.851 | 1 |
| d2,X | d3,X | −15.291 | −0.025 | 15.240 | 1 |
| d2,X | d4,X | −14.183 | 1.083 | 16.349 | 1 |
| d2,X | d5,X | −15.189 | 0.077 | 15.342 | 1 |
| d2,X | d6,X | −17.881 | −2.615 | 12.651 | 1 |
| d3,X | d4,X | −14.437 | 1.108 | 16.654 | 1 |
| d3,X | d5,X | −15.444 | 0.102 | 15.648 | 1 |
| d3,X | d6,X | −18.135 | −2.589 | 12.956 | 1 |
| d4,X | d5,X | −16.552 | −1.006 | 14.540 | 1 |
| d4,X | d6,X | −19.244 | −3.698 | 11.848 | 1 |
| d5,X | d6,X | −18.237 | −2.691 | 12.854 | 1 |
| d1,S | d2,S | −17.996 | −3.015 | 11.965 | 1 |
| d1,S | d3,S | −15.862 | −0.597 | 14.669 | 1 |
| d1,S | d4,S | −15.753 | −0.488 | 14.778 | 1 |
| d1,S | d5,S | −12.589 | 2.677 | 17.942 | 1 |
| d1,S | d6,S | −9.620 | 5.646 | 20.911 | 0.998 |
| d2,S | d3,S | −12.847 | 2.419 | 17.685 | 1 |
| d2,S | d4,S | −12.738 | 2.528 | 17.794 | 1 |
| d2,S | d5,S | −9.574 | 5.692 | 20.958 | 0.998 |
| d2,S | d6,S | −6.605 | 8.661 | 23.927 | 0.880 |
| d3,S | d4,S | −15.437 | 0.109 | 15.655 | 1 |
| d3,S | d5,S | −12.273 | 3.273 | 18.819 | 1 |
| d3,S | d6,S | −9.304 | 6.242 | 21.788 | 0.996 |
| d4,S | d5,S | −12.382 | 3.164 | 18.710 | 1 |
| d4,S | d6,S | −9.413 | 6.133 | 21.679 | 0.997 |
| d5,S | d6,S | −12.577 | 2.969 | 18.515 | 1 |

**Appendix 1—table 35.** Two-way ANOVA for effect of imaging day and valence of odor on relative eye size during cue presentation (*Figure 2—figure supplement 5G*).

| Source | Sum Sq. | d.f. | Mean Sq. | F | Prob >F |
|---|---|---|---|---|---|
| days | 0.293 | 5 | 0.059 | 12.301 | 8.84E-11 |
| valence | 0.014 | 2 | 0.007 | 1.523 | 0.220 |
| days:valence | 0.121 | 10 | 0.012 | 2.546 | 5.95E-03 |
| Error | 1.313 | 276 | 0.005 | | |
| Total | 1.739 | 293 | | | |

**Appendix 1—table 36.** Two-way ANOVA for effect of imaging day and valence of odor on relative eye size during unconditioned stimulus (*Figure 2—figure supplement 5G*).

| Source | Sum Sq. | d.f. | Mean Sq. | F | Prob >F |
|---|---|---|---|---|---|
| days | 0.178 | 5 | 0.036 | 4.529 | 5.51E-04 |
| valence | 0.040 | 2 | 0.020 | 2.574 | 0.078 |
| days:valence | 0.167 | 10 | 0.017 | 2.123 | 2.29E-02 |
| Error | 2.167 | 276 | 0.008 | | |
| Total | 2.547 | 293 | | | |

**Appendix 1—table 37.** Four-way ANOVA for effect of imaging day, valence, functional group, and region on the percentage of neurons responsive to a given odor (*Figure 2—figure supplement 7A*).

| Source | Sum Sq. | d.f. | Mean Sq. | F | Prob >F |
|---|---|---|---|---|---|
| ket | 0.051 | 1 | 0.051 | 3.828 | 0.051 |
| val. | 0.787 | 2 | 0.393 | 29.766 | 2.48E-12 |
| reg. | 6.62E-03 | 2 | 0.003 | 0.250 | 0.779 |
| day | 0.928 | 2 | 0.464 | 35.101 | 3.57E-14 |
| ket:val. | 0.024 | 2 | 0.012 | 0.911 | 0.403 |
| ket:reg. | 0.018 | 2 | 0.009 | 0.671 | 0.512 |
| ket:day | 0.057 | 2 | 0.029 | 2.163 | 0.117 |
| val.:reg. | 0.622 | 4 | 0.155 | 11.763 | 8.94E-09 |
| val.:day | 0.264 | 4 | 0.066 | 4.995 | 6.85E-04 |
| reg.:day | 0.328 | 4 | 0.082 | 6.212 | 8.79E-05 |
| ket:val.:reg. | 0.054 | 4 | 0.014 | 1.025 | 0.395 |
| ket:val.:day | 0.095 | 4 | 0.024 | 1.796 | 0.130 |
| ket:reg.:day | 0.014 | 4 | 0.004 | 0.272 | 0.896 |
| val.:reg.:day | 0.241 | 8 | 0.030 | 2.277 | 0.023 |
| ket:val.:reg.: day | 0.062 | 8 | 0.008 | 0.582 | 0.792 |
| Error | 3.331 | 252 | 0.013 | | |
| Total | 6.670 | 305 | | | |

**Appendix 1—table 38.** Linear model of the fixed effects of region, imaging day, and valence and the random effect of individual animal on |ΔΔF/F| (*Figure 2—figure supplement 8A*).

**Formula:**

| Fmag ~1 + reg*day +reg*val +day*val +reg:day:val + (1 | id) |
|---|

**Model information**

| # of observations: | Fixed effects coefficients: | Random effect coefficients: | Covariance parameters: |
|---|---|---|---|
| 11,160 | 12 | 17 | 2 |

**Model fit statistics:**

| AIC | BIC | Log Likelihood | Deviance |
|---|---|---|---|
| 9050.1 | 9152.6 | –4511.1 | 9022.1 |

**Fixed effects coefficients (95% CIs):**

| Name | Estimate | SE | tStat | DF | pValue |
|---|---|---|---|---|---|

*Appendix 1—table 38 Continued on next page*

*Appendix 1—table 38 Continued*

**Formula:**

| intercept | 0.469 | 0.0155 | 30.228 | 11,148 | 5.43E-193 |
|---|---|---|---|---|---|
| reg_D1 | –0.0222 | 0.0205 | –1.0829 | 11,148 | 0.279 |
| reg_VP | 0.0192 | 0.0255 | 0.753 | 11,148 | 0.452 |
| day | –0.0140 | 0.00707 | –1.973 | 11,148 | 0.0485 |
| val | –0.0244 | 0.0190 | –1.286 | 11,148 | 0.198 |
| reg_D1:day | 0.00806 | 0.00947 | 0.852 | 11,148 | 0.394 |
| reg_VP:day | –0.0101 | 0.0117 | –0.862 | 11,148 | 0.389 |
| reg_D1:val | –0.00659 | 0.0251 | –0.262 | 11,148 | 0.793 |
| reg_VP:val | –0.0359 | 0.0312 | –1.150 | 11,148 | 0.250 |
| day:val | 0.00799 | 0.00866 | 0.922 | 11,148 | 0.356 |
| reg_D1:day:val | 0.0294 | 0.0116 | 2.539 | 11,148 | 0.0111 |
| reg_VP:day:val | 0.0826 | 0.0143 | 5.762 | 11,148 | 8.5E-9 |

**Appendix 1—table 39.** Linear model of the fixed effects of region and imaging day, and the random effect of individual animals on the auROC of single-neuron {S vs. X|P} classifiers (*Figure 3—figure supplement 1F*).

**Formula:**

auROC {S vs. X|P}~1 + region*day + (1 | id)

Model information

| Number of observations: | Fixed effects coefficients: | Random effect coefficients: | Covariance parameters: |
|---|---|---|---|
| 1860 | 6 | 17 | 2 |

Model fit statistics:

| AIC | BIC | Log Likelihood | Deviance |
|---|---|---|---|
| –4303.7 | –4259.5 | 2159.8 | –4319.7 |

Fixed effects coefficients (95% CIs):

| Name | Estimate | SE | tStat | DF | pValue |
|---|---|---|---|---|---|
| intercept | 0.617 | 0.009 | 68.007 | 1854 | 0 |
| reg_D1 | –0.002 | 0.012 | –0.186 | 1854 | 0.853 |
| reg_VP | –0.037 | 0.014 | –2.665 | 1854 | 7.76E-03 |
| day | 0.003 | 0.001 | 1.874 | 1854 | 0.061 |
| reg_D1:day | 0.007 | 0.002 | 3.484 | 1854 | 5.06E-04 |
| reg_VP:day | 0.029 | 0.002 | 12.056 | 1854 | 2.80E-32 |

**Appendix 1—table 40.** Two-way ANOVA for effect of imaging day and region on the median auROC value of {S vs. X|P} classifiers for each animal (*Figure 3—figure supplement 1F*).

| Source | Sum Sq. | d.f. | Mean Sq. | F | Prob >F |
|---|---|---|---|---|---|
| region | 0.030 | 2 | 0.015 | 14.686 | 1.46E-05 |
| day | 0.050 | 2 | 0.025 | 24.336 | 9.58E-08 |
| region:day | 0.041 | 4 | 0.010 | 10.001 | 8.88E-06 |
| Error | 0.043 | 42 | 0.001 | | |
| Total | 0.158 | 50 | | | |

**Appendix 1—table 41.** Post hoc comparison of the median auROC value for {S vs. X|P} across imaging day and region (*Figure 3—figure supplement 1F*).

| Group A | Group B | Lower Limit | A-B | Upper Limit | p-value |
|---------|---------|-------------|------|-------------|---------|
| D2 OT,d1 | D2 OT,d3 | –0.072 | –0.012 | 0.049 | 0.999 |
| D2 OT,d1 | D2 OT,d6 | –0.075 | –0.014 | 0.046 | 0.997 |
| D2 OT,d3 | D2 OT,d6 | –0.063 | –0.003 | 0.058 | 1 |
| D1 OT,d3 | D1 OT,d6 | –0.072 | –0.012 | 0.049 | 0.999 |
| D1 OT,d1 | D1 OT,d3 | –0.091 | –0.030 | 0.030 | 0.779 |
| D1 OT,d1 | D1 OT,d6 | –0.102 | –0.042 | 0.019 | 0.387 |
| VP,d1 | VP,d3 | –0.130 | –0.064 | 0.002 | 6.62E-02 |
| VP,d1 | VP,d6 | –0.241 | –0.174 | –0.108 | 2.83E-09 |
| VP,d3 | VP,d6 | –0.177 | –0.111 | –0.044 | 7.91E-05 |
| D2 OT,d1 | D1 OT,d1 | –0.056 | 0.005 | 0.065 | 1 |
| D2 OT,d1 | VP,d1 | –0.050 | 0.013 | 0.076 | 0.999 |
| D1 OT,d1 | VP,d1 | –0.055 | 0.008 | 0.072 | 1 |
| D2 OT,d3 | D1 OT,d3 | –0.074 | –0.014 | 0.047 | 0.998 |
| D2 OT,d3 | VP,d3 | –0.103 | –0.039 | 0.024 | 0.537 |
| D1 OT,d3 | VP,d3 | –0.089 | –0.026 | 0.038 | 0.921 |
| D2 OT,d6 | D1 OT,d6 | –0.083 | –0.023 | 0.038 | 0.948 |
| D2 OT,d6 | VP,d6 | –0.210 | –0.147 | –0.084 | 7.68E-08 |
| D1 OT,d6 | VP,d6 | –0.188 | –0.124 | –0.061 | 3.44E-06 |

**Appendix 1—table 42.** Linear model of the fixed effects of region and imaging day, and the random effect of individual animals on the auROC of single-neuron {$S_K$ vs. $S_T$} classifiers (*Figure 3—figure supplement 1G*).

**Formula:**

auROC {$S_K$ vs. $S_T$}~1 + region*day + (1 | id)

Model information

| Number of observations: | Fixed effects coefficients: | Random effect coefficients: | Covariance parameters: |
|---|---|---|---|
| 1860 | 6 | 17 | 2 |

Model fit statistics:

| AIC | BIC | Log Likelihood | Deviance |
|---|---|---|---|
| –2908.8 | –2864.5 | 1462.4 | –2924.8 |

Fixed effects coefficients (95% CIs):

| Name | Estimate | SE | tStat | DF | pValue |
|------|----------|-----|-------|-----|--------|
| intercept | 0.636 | 0.015 | 42.040 | 1854 | 7.72E-272 |
| region_D1 | –0.020 | 0.021 | –0.962 | 1854 | 0.336 |
| region_VP | –0.024 | 0.024 | –1.014 | 1854 | 0.311 |
| day | 9.67E-04 | 5.25E-03 | 0.184 | 1854 | 0.854 |
| region_D1:day | 0.011 | 0.007 | 1.622 | 1854 | 0.105 |
| region_VP:day | –0.003 | 0.009 | –0.346 | 1854 | 0.730 |

**Appendix 1—table 43.** Two-way ANOVA for effect of imaging day and region on the median auROC value of {$S_K$ vs. $S_T$} classifiers for each animal (*Figure 3—figure supplement 1G*).

| Source | Sum Sq. | d.f. | Mean Sq. | F | Prob >F |
|---|---|---|---|---|---|
| region | 0.011 | 2 | 5.54E-03 | 2.793 | 0.073 |
| day | 2.91E-04 | 2 | 1.46E-04 | 0.073 | 0.929 |
| region:day | 3.66E-03 | 4 | 9.16E-04 | 0.462 | 0.763 |
| Error | 0.083 | 42 | 1.98E-03 | | |
| Total | 0.098 | 50 | | | |

**Appendix 1—table 44.** Linear model of the fixed effects of region and imaging day, and the random effect of individual animals on the single-neuron valence scores (*Figure 3—figure supplement 1H*).

**Formula:**

| score ~1 + region*day + (1 | id) |
|---|

Model information

| Number of observations: | Fixed effects coefficients: | Random effect coefficients: | Covariance parameters: |
|---|---|---|---|
| 1860 | 6 | 17 | 2 |

Model fit statistics:

| AIC | BIC | Log Likelihood | Deviance |
|---|---|---|---|
| −3219.9 | −3175.7 | 1618 | −3235.9 |

Fixed effects coefficients (95% CIs):

| Name | Estimate | SE | tStat | DF | pValue |
|---|---|---|---|---|---|
| intercept | −0.023 | 0.015 | −1.563 | 1854 | 0.118 |
| region_D1 | 4.19E-03 | 0.020 | 0.204 | 1854 | 0.838 |
| region_VP | −0.062 | 0.023 | −2.648 | 1854 | 8.16E-03 |
| day | 5.84E-03 | 4.83E-03 | 1.209 | 1854 | 0.227 |
| region_D1:day | 6.40E-03 | 6.45E-03 | 0.991 | 1854 | 0.322 |
| region_VP:day | 0.075 | 7.98E-03 | 9.384 | 1854 | 1.80E-20 |

**Appendix 1—table 45.** Two-way ANOVA for effect of imaging day and region on the median valence score for each animal (*Figure 3—figure supplement 1H*).

| Source | Sum Sq. | d.f. | Mean Sq. | F | Prob >F |
|---|---|---|---|---|---|
| reg | 0.070 | 2 | 0.035 | 13.163 | 3.65E-05 |
| day | 0.032 | 2 | 0.016 | 6.073 | 4.82E-03 |
| reg:day | 0.045 | 4 | 0.011 | 4.264 | 5.50E-03 |
| Error | 0.111 | 42 | 2.65E-03 | | |
| Total | 0.253 | 50 | | | |

**Appendix 1—table 46.** Post hoc comparison of the median valence scores across imaging day and region (*Figure 3—figure supplement 1H*).

| Group A | Group B | Lower Limit | A-B | Upper Limit | p-value |
|---|---|---|---|---|---|
| D2 OT,d1 | D2 OT,d3 | −0.100 | −0.003 | 0.094 | 1 |
| D2 OT,d1 | D2 OT,d6 | −0.109 | −0.012 | 0.086 | 1 |

*Appendix 1—table 46 Continued on next page*

*Appendix 1—table 46 Continued*

| Group A | Group B | Lower Limit | A-B | Upper Limit | p-value |
|---------|---------|-------------|-----|-------------|---------|
| D2 OT,d3 | D2 OT,d6 | –0.106 | –0.008 | 0.089 | 1 |
| D1 OT,d1 | D1 OT,d3 | –0.100 | –0.002 | 0.095 | 1 |
| D1 OT,d1 | D1 OT,d6 | –0.102 | –0.005 | 0.092 | 1 |
| D1 OT,d3 | D1 OT,d6 | –0.100 | –0.003 | 0.095 | 1 |
| VP,d1 | VP,d3 | –0.157 | –0.051 | 0.056 | 0.819 |
| VP,d1 | VP,d6 | –0.271 | –0.165 | –0.058 | 2.87E-04 |
| VP,d3 | VP,d6 | –0.220 | –0.114 | –0.007 | 2.87E-02 |
| D2 OT,d1 | D1 OT,d1 | –0.112 | –0.015 | 0.082 | 1 |
| D2 OT,d1 | VP,d1 | –0.122 | –0.020 | 0.082 | 1 |
| D1 OT,d1 | VP,d1 | –0.107 | –0.005 | 0.097 | 1 |
| D2 OT,d3 | D1 OT,d3 | –0.111 | –0.014 | 0.083 | 1 |
| D2 OT,d3 | VP,d3 | –0.169 | –0.067 | 0.034 | 0.447 |
| D1 OT,d3 | VP,d3 | –0.155 | –0.053 | 0.049 | 0.736 |
| D2 OT,d6 | D1 OT,d6 | –0.105 | –0.008 | 0.089 | 1 |
| D2 OT,d6 | VP,d6 | –0.275 | –0.173 | –0.071 | 6.07E-05 |
| D1 OT,d6 | VP,d6 | –0.266 | –0.164 | –0.062 | 1.41E-04 |

**Appendix 1—table 47.** Two-way ANOVA for effect of imaging day and region on the median single-neuron MNR accuracy for each animal *Figure 3—figure supplement 2B*.

| Source | Sum Sq. | d.f. | Mean Sq. | F | Prob >F |
|--------|---------|------|----------|---|---------|
| reg | 0.003 | 2 | 0.002 | 4.286 | 2.02E-02 |
| day | 0.006 | 2 | 0.003 | 7.181 | 2.08E-03 |
| reg:day | 0.004 | 4 | 0.001 | 2.622 | 4.82E-02 |
| Error | 0.017 | 42 | 0.000 | | |
| Total | 0.030 | 50 | | | |

**Appendix 1—table 48.** Post hoc comparison of median single-neuron MNR accuracy across imaging day and region (*Figure 3—figure supplement 2B*).

| Group A | Group B | Lower Limit | A-B | Upper Limit | p-value |
|---------|---------|-------------|-----|-------------|---------|
| D2 OT,d1 | D2 OT,d3 | –0.051 | –0.013 | 0.024 | 0.960 |
| D2 OT,d1 | D2 OT,d6 | –0.045 | –0.007 | 0.031 | 0.999 |
| D2 OT,d3 | D2 OT,d6 | –0.032 | 0.006 | 0.044 | 1 |
| D1 OT,d1 | D1 OT,d3 | –0.047 | –0.010 | 0.028 | 0.996 |
| D1 OT,d1 | D1 OT,d6 | –0.050 | –0.012 | 0.026 | 0.981 |
| D1 OT,d3 | D1 OT,d6 | –0.040 | –0.002 | 0.036 | 1 |
| VP,d1 | VP,d3 | –0.073 | –0.031 | 0.011 | 0.294 |
| VP,d1 | VP,d6 | –0.099 | –0.058 | –0.016 | 1.47E-03 |
| VP,d3 | VP,d6 | –0.068 | –0.027 | 0.015 | 0.490 |
| D2 OT,d1 | D1 OT,d1 | –0.052 | –0.014 | 0.024 | 0.953 |
| D2 OT,d1 | VP,d1 | –0.037 | 0.003 | 0.043 | 1 |

*Appendix 1—table 48 Continued on next page*

*Appendix 1—table 48 Continued*

| Group A | Group B | Lower Limit | A-B | Upper Limit | p-value |
|---------|---------|-------------|-----|-------------|---------|
| D1 OT,d1 | VP,d1 | −0.023 | 0.017 | 0.057 | 0.896 |
| D2 OT,d3 | D1 OT,d3 | −0.048 | −0.010 | 0.028 | 0.994 |
| D2 OT,d3 | VP,d3 | −0.054 | −0.014 | 0.025 | 0.955 |
| D1 OT,d3 | VP,d3 | −0.044 | −0.005 | 0.035 | 1 |
| D2 OT,d6 | D1 OT,d6 | −0.057 | −0.019 | 0.019 | 0.797 |
| D2 OT,d6 | VP,d6 | −0.087 | −0.047 | −0.008 | 9.47E-03 |
| D1 OT,d6 | VP,d6 | −0.069 | −0.029 | 0.011 | 0.329 |

**Appendix 1—table 49.** Two-way ANOVA for effect of imaging day and region on the median single-neuron MNR shuffled accuracy for each animal *Figure 3—figure supplement 2B*.

| Source | Sum Sq. | d.f. | Mean Sq. | F | Prob >F |
|--------|---------|------|----------|---|---------|
| reg | 2.04E-08 | 2 | 1.02E-08 | 0.040 | 0.961 |
| day | 7.71E-07 | 2 | 3.86E-07 | 1.519 | 0.231 |
| reg:day | 1.22E-06 | 4 | 3.05E-07 | 1.200 | 0.325 |
| Error | 1.07E-05 | 42 | 2.54E-07 | | |
| Total | 1.26E-05 | 50 | | | |

**Appendix 1—table 50.** Two-way ANOVA for effect of imaging day and region on the median S-cue/S-cue confusion for each animal *Figure 3—figure supplement 2D*.

| Source | Sum Sq. | d.f. | Mean Sq. | F | Prob >F |
|--------|---------|------|----------|---|---------|
| reg | 0.074 | 2 | 0.037 | 27.720 | 2.11E-08 |
| day | 0.019 | 2 | 9.43E-03 | 7.066 | 2.26E-03 |
| reg:day | 0.011 | 4 | 2.65E-03 | 1.986 | 1.14E-01 |
| Error | 0.056 | 42 | 1.33E-03 | | |
| Total | 0.157 | 50 | | | |

**Appendix 1—table 51.** Post hoc comparison of median S-cue/S-cue confusion across imaging day and region *Figure 3—figure supplement 2D*.

| Group A | Group B | Lower Limit | A-B | Upper Limit | p-value |
|---------|---------|-------------|-----|-------------|---------|
| D2 OT,d1 | D1 OT,d1 | −0.090 | −0.021 | 0.048 | 0.985 |
| D2 OT,d1 | VP,d1 | −0.132 | −0.060 | 0.012 | 0.174 |
| D1 OT,d1 | VP,d1 | −0.111 | −0.039 | 0.033 | 0.700 |
| D2 OT,d3 | D1 OT,d3 | −0.095 | −0.026 | 0.043 | 0.940 |
| D2 OT,d3 | VP,d3 | −0.156 | −0.084 | −0.011 | 1.30E-02 |
| D1 OT,d3 | VP,d3 | −0.130 | −0.057 | 0.015 | 0.223 |
| D2 OT,d6 | D1 OT,d6 | −0.084 | −0.015 | 0.054 | 0.998 |
| D2 OT,d6 | VP,d6 | −0.204 | −0.132 | −0.059 | 1.55E-05 |
| D1 OT,d6 | VP,d6 | −0.189 | −0.116 | −0.044 | 1.46E-04 |
| D2 OT,d1 | D2 OT,d3 | −0.079 | −0.010 | 0.059 | 1 |
| D2 OT,d1 | D2 OT,d6 | −0.094 | −0.025 | 0.044 | 0.955 |
| D2 OT,d3 | D2 OT,d6 | −0.084 | −0.015 | 0.054 | 0.998 |

*Appendix 1—table 51 Continued on next page*

*Appendix 1—table 51 Continued*

| Group A | Group B | Lower Limit | A-B | Upper Limit | p-value |
|---|---|---|---|---|---|
| D1 OT,d1 | D1 OT,d3 | –0.084 | –0.015 | 0.054 | 0.998 |
| D1 OT,d1 | D1 OT,d6 | –0.088 | –0.019 | 0.049 | 0.990 |
| D1 OT,d3 | D1 OT,d6 | –0.073 | –0.004 | 0.065 | 1 |
| VP,d1 | VP,d3 | –0.109 | –0.033 | 0.042 | 0.874 |
| VP,d1 | VP,d6 | –0.172 | –0.097 | –0.021 | 4.11E-03 |
| VP,d3 | VP,d6 | –0.139 | –0.063 | 0.012 | 0.165 |

**Appendix 1—table 52.** Two-way ANOVA for effect of imaging day and region on the median confusion within functional groups for each animal *Figure 3—figure supplement 2E*.

| Source | Sum Sq. | d.f. | Mean Sq. | F | Prob >F |
|---|---|---|---|---|---|
| reg | 1.60E-03 | 2 | 7.99E-04 | 0.681 | 0.512 |
| day | 0.017 | 2 | 8.71E-03 | 7.426 | 1.73E-03 |
| reg:day | 2.53E-03 | 4 | 6.31E-04 | 0.538 | 0.708 |
| Error | 0.049 | 42 | 1.17E-03 | | |
| Total | 0.070 | 50 | | | |

**Appendix 1—table 53.** Post hoc comparison of median within-function group confusion across imaging day and region *Figure 3—figure supplement 2E*.

| Group A | Group B | Lower Limit | A-B | Upper Limit | p-value |
|---|---|---|---|---|---|
| D2 OT,d1 | D2 OT,d3 | –0.055 | 0.009 | 0.074 | 1.000 |
| D2 OT,d1 | D2 OT,d6 | –0.033 | 0.031 | 0.096 | 0.804 |
| D2 OT,d3 | D2 OT,d6 | –0.042 | 0.022 | 0.087 | 0.967 |
| D1 OT,d1 | D1 OT,d3 | –0.063 | 0.002 | 0.066 | 1 |
| D1 OT,d1 | D1 OT,d6 | –0.034 | 0.031 | 0.095 | 0.828 |
| D1 OT,d3 | D1 OT,d6 | –0.036 | 0.029 | 0.093 | 0.871 |
| VP,d1 | VP,d3 | –0.060 | 0.011 | 0.082 | 1 |
| VP,d1 | VP,d6 | –0.005 | 0.066 | 0.136 | 0.089 |
| VP,d3 | VP,d6 | –0.016 | 0.054 | 0.125 | 0.255 |
| D2 OT,d1 | D1 OT,d1 | –0.061 | 0.004 | 0.068 | 1 |
| D2 OT,d1 | VP,d1 | –0.067 | 0.001 | 0.069 | 1 |
| D1 OT,d1 | VP,d1 | –0.071 | –0.003 | 0.065 | 1 |
| D2 OT,d3 | D1 OT,d3 | –0.068 | –0.004 | 0.061 | 1 |
| D2 OT,d3 | VP,d3 | –0.065 | 0.003 | 0.070 | 1 |
| D1 OT,d3 | VP,d3 | –0.061 | 0.006 | 0.074 | 1 |
| D2 OT,d6 | D1 OT,d6 | –0.062 | 0.003 | 0.067 | 1 |
| D2 OT,d6 | VP,d6 | –0.033 | 0.035 | 0.103 | 0.756 |
| D1 OT,d6 | VP,d6 | –0.036 | 0.032 | 0.100 | 0.828 |

**Appendix 1—table 54.** Two-way ANOVA for effect of imaging day and region on the mean accuracy for linear classification of {S vs. X} using population data (*Figure 5—figure supplement 1B*).

| Source | Sum Sq. | d.f. | Mean Sq. | F | Prob >F |
|---|---|---|---|---|---|
| region | 0.010 | 2 | 4.92E-03 | 0.868 | 0.427 |
| day | 0.178 | 2 | 0.089 | 15.734 | 7.95E-06 |
| region:day | 0.064 | 4 | 0.016 | 2.846 | 3.56E-02 |
| Error | 0.238 | 42 | 5.67E-03 | | |
| Total | 0.476 | 50 | | | |

**Appendix 1—table 55.** Post hoc comparison of mean {S vs. X} accuracy across imaging day and region (*Figure 5—figure supplement 1B*).

| Group A | Group B | Lower Limit | A-B | Upper Limit | p-value |
|---|---|---|---|---|---|
| D1 OT,d1 | D2 OT,d1 | –0.108 | 0.034 | 0.176 | 0.997 |
| D1 OT,d1 | VP,d1 | –0.018 | 0.131 | 0.280 | 0.127 |
| D2 OT,d1 | VP,d1 | –0.052 | 0.097 | 0.246 | 0.476 |
| D1 OT,d3 | D2 OT,d3 | –0.081 | 0.061 | 0.203 | 0.889 |
| D1 OT,d3 | VP,d3 | –0.153 | –0.004 | 0.145 | 1 |
| D2 OT,d3 | VP,d3 | –0.214 | –0.065 | 0.084 | 0.880 |
| D1 OT,d6 | D2 OT,d6 | –0.142 | 0.000 | 0.142 | 1 |
| D1 OT,d6 | VP,d6 | –0.202 | –0.053 | 0.096 | 0.960 |
| D2 OT,d6 | VP,d6 | –0.202 | –0.053 | 0.096 | 0.960 |
| D1 OT,d1 | D1 OT,d3 | –0.190 | –0.048 | 0.094 | 0.971 |
| D1 OT,d1 | D1 OT,d6 | –0.214 | –0.072 | 0.070 | 0.765 |
| D1 OT,d3 | D1 OT,d6 | –0.166 | –0.024 | 0.118 | 1 |
| D2 OT,d1 | D2 OT,d3 | –0.163 | –0.021 | 0.121 | 1 |
| D2 OT,d1 | D2 OT,d6 | –0.248 | –0.106 | 0.036 | 0.288 |
| D2 OT,d3 | D2 OT,d6 | –0.227 | –0.085 | 0.057 | 0.575 |
| VP,d1 | VP,d3 | –0.338 | –0.183 | –0.027 | 1.12E-02 |
| VP,d1 | VP,d6 | –0.411 | –0.256 | –0.100 | 1.02E-04 |
| VP,d3 | VP,d6 | –0.229 | –0.073 | 0.082 | 0.830 |

**Appendix 1—table 56.** Two-way ANOVA for effect of imaging day and region on the mean accuracy for linear classification of {S vs. P} using population data (Fig5-1C).

| Source | Sum Sq. | d.f. | Mean Sq. | F | Prob >F |
|---|---|---|---|---|---|
| region | 0.020 | 2 | 0.010 | 1.776 | 0.182 |
| day | 0.255 | 2 | 0.128 | 22.389 | 2.41E-07 |
| region:day | 0.042 | 4 | 0.010 | 1.822 | 0.143 |
| Error | 0.240 | 42 | 5.71E-03 | | |
| Total | 0.543 | 50 | | | |

**Appendix 1—table 57.** Post hoc comparison of mean {S vs. P} accuracy across imaging day and region (*Figure 5—figure supplement 1C*).

| Group A | Group B | Lower Limit | A-B | Upper Limit | p-value |
|---|---|---|---|---|---|
| D1 OT,d1 | D2 OT,d1 | –0.104 | 0.038 | 0.181 | 0.993 |
| D1 OT,d1 | VP,d1 | –0.051 | 0.098 | 0.248 | 0.453 |
| D2 OT,d1 | VP,d1 | –0.089 | 0.060 | 0.210 | 0.920 |
| D1 OT,d3 | D2 OT,d3 | –0.057 | 0.085 | 0.228 | 0.579 |
| D1 OT,d3 | VP,d3 | –0.134 | 0.015 | 0.165 | 1 |
| D2 OT,d3 | VP,d3 | –0.219 | –0.070 | 0.079 | 0.835 |
| D1 OT,d6 | D2 OT,d6 | –0.124 | 0.019 | 0.161 | 1 |
| D1 OT,d6 | VP,d6 | –0.192 | –0.043 | 0.107 | 0.989 |
| D2 OT,d6 | VP,d6 | –0.211 | –0.062 | 0.088 | 0.910 |
| D1 OT,d1 | D1 OT,d3 | –0.233 | –0.090 | 0.052 | 0.506 |
| D1 OT,d1 | D1 OT,d6 | –0.262 | –0.119 | 0.023 | 0.165 |
| D1 OT,d3 | D1 OT,d6 | –0.172 | –0.029 | 0.113 | 0.999 |
| D2 OT,d1 | D2 OT,d3 | –0.186 | –0.043 | 0.099 | 0.985 |
| D2 OT,d1 | D2 OT,d6 | –0.281 | –0.139 | 0.004 | 0.061 |
| D2 OT,d3 | D2 OT,d6 | –0.238 | –0.096 | 0.047 | 0.426 |
| VP,d1 | VP,d3 | –0.329 | –0.173 | –0.017 | 1.97E-02 |
| VP,d1 | VP,d6 | –0.417 | –0.261 | –0.105 | 7.71E-05 |
| VP,d3 | VP,d6 | –0.244 | –0.088 | 0.069 | 0.662 |

**Appendix 1—table 58.** Two-way ANOVA for effect of imaging day and region on the accuracy for linear classification of {$S_K$ vs. $S_T$} using population data (*Figure 5—figure supplement 1D*).

| Source | Sum Sq. | d.f. | Mean Sq. | F | Prob >F |
|---|---|---|---|---|---|
| region | 0.085 | 2 | 0.042 | 3.600 | 0.036 |
| day | 0.033 | 2 | 0.017 | 1.415 | 0.254 |
| region:day | 0.042 | 4 | 0.010 | 0.890 | 0.478 |
| Error | 0.493 | 42 | 0.012 | | |
| Total | 0.655 | 50 | | | |

**Appendix 1—table 59.** Post hoc comparison of {$S_K$ vs. $S_T$} accuracy across imaging day and region (*Figure 5—figure supplement 1D*).

| Group A | Group B | Lower Limit | A-B | Upper Limit | p-value |
|---|---|---|---|---|---|
| D1 OT,d1 | D2 OT,d1 | –0.202 | 0.003 | 0.207 | 1 |
| D1 OT,d1 | VP,d1 | –0.121 | 0.094 | 0.308 | 0.879 |
| D2 OT,d1 | VP,d1 | –0.123 | 0.091 | 0.306 | 0.896 |
| D1 OT,d3 | D2 OT,d3 | –0.238 | –0.033 | 0.171 | 1 |
| D1 OT,d3 | VP,d3 | –0.216 | –0.002 | 0.213 | 1 |
| D2 OT,d3 | VP,d3 | –0.183 | 0.032 | 0.246 | 1 |
| D1 OT,d6 | D2 OT,d6 | –0.307 | –0.103 | 0.102 | 0.776 |
| D1 OT,d6 | VP,d6 | –0.135 | 0.079 | 0.294 | 0.950 |
| D2 OT,d6 | VP,d6 | –0.032 | 0.182 | 0.397 | 0.153 |

*Appendix 1—table 59 Continued on next page*

*Appendix 1—table 59 Continued*

| Group A | Group B | Lower Limit | A-B | Upper Limit | p-value |
|---|---|---|---|---|---|
| D1 OT,d1 | D1 OT,d3 | –0.199 | 0.006 | 0.210 | 1 |
| D1 OT,d1 | D1 OT,d6 | –0.227 | –0.022 | 0.182 | 1 |
| D1 OT,d3 | D1 OT,d6 | –0.232 | –0.028 | 0.177 | 1 |
| D2 OT,d1 | D2 OT,d3 | –0.235 | –0.031 | 0.174 | 1 |
| D2 OT,d1 | D2 OT,d6 | –0.332 | –0.128 | 0.077 | 0.525 |
| D2 OT,d3 | D2 OT,d6 | –0.302 | –0.097 | 0.107 | 0.823 |
| VP,d1 | VP,d3 | –0.314 | –0.090 | 0.134 | 0.922 |
| VP,d1 | VP,d6 | –0.261 | –0.037 | 0.187 | 1 |
| VP,d3 | VP,d6 | –0.171 | 0.053 | 0.277 | 0.997 |

