## [Editor Report · eLife assessment]

This **important** study by Lee and colleagues examined how neural representations are transformed between the olfactory tubercle (OT) and the ventral pallidum (VP) using single neuron calcium imaging in head-fixed mice trained in classical conditioning. They show that the dimensionality of neural responses is lower in the VP than in the OT and suggest that VP responses represent values in a more abstract form at the single neuron level while OT contains more odor information, potentially enhancing odor contrast. The results are overall **convincing** and this study provides insights into how odor information is transformed in the olfactory system.

---

## [Referee Report · Reviewer #1 (Public Review)]

In this manuscript, Lee et al. compared encoding of odor identity and value by calcium signaling from neurons in the ventral pallidum (VP) in comparison to D1 and D2 neurons in the olfactory tubercle (OT).

Strengths:

They utilize a strong comparative approach, which allows the comparison of signals in two directly connected regions. First, they demonstrate that both D1 and D2 OT neurons project strongly to the VP, but not the VTA or other examined regions, in contrast to accumbal D1 neurons which project strongly to the VTA as well as the VP. They examine single unit calcium activity in a robust olfactory cue conditioning paradigm that allows them to differentiate encoding of olfactory identity versus value, by incorporating two different sucrose, neutral and air puff cues with different chemical characteristics. They then use multiple analytical approaches to demonstrate strong, low-dimensional encoding of cue value in the VP, and more robust, high-dimensional encoding of odor identity by both D1 and D2 OT neurons, though D1 OT neurons are still somewhat modulated by reward contingency/value. Finally, they utilize a modified conditioning paradigm that dissociates reward probability and lick vigor to demonstrate that VP encoding of cue value is not dependent on encoding of lick vigor during sucrose cues, and that separable populations of VP neuros encode cue value/sucrose probability and lick vigor. Direct comparisons of single unit responses between the two regions now utilize linear mixed effects models with random effects for subject,

Weaknesses:

The manuscript still includes mention of differences in effect size or differing "levels" of significance between VP and OT D1 neurons without reports of a direct comparisons between the two populations. This is somewhat mitigated by the comprehensive statistical reporting in the supplemental information, but interpretation of some of these results is clouded by the inclusion of OT D2 neurons in these analyses, and the limited description or contextualization in the main text.

---

## [Referee Report · Reviewer #2 (Public Review)]

We appreciate the authors revision of this manuscript and toning down some of the statements regarding "contradictory" results. We still have some concerns about the major claims of this paper which lead us to suggest this paper undergo more revision as follows since, in its present form, we fear this paper is misleading for the field in two areas. here is a brief outline:

(1) Despite acknowledging that the injections only occurred in the anteromedial aspect of the tubercle, the authors still assert broad conclusions regarding where the tubercle projects and what the tubercle does. for instance, even the abstract states "both D1 and D2 neurons of the OT project primarily to the VP and minimally elsewhere" without mention that this is the "anteromedial OT". Every conclusion needs to specify this is stemming from evidence in just the anteromedial tubercle, as the authors do in some parts of the the discussion.

(2) The authors now frame the 2P imaging data that D1 neuron activity reflects "increased contrast of identity or an intermediate and multiplexed encoding of valence and identity". I struggle to understand what the authors are actually concluding here. Later in discussion, the authors state that they saw that OT D1 and D2 neurons "encode odor valence" (line 510). We appreciate the authors note that there is "poor standardization" when it comes to defining valence (line 521). We are ok with the authors speculating and think this revision is more forthcoming regarding the results and better caveats the conclusions. I suggest in abstract the authors adjust line 14/15 to conclude that, "While D1 OT neurons showed larger responses to rewarded odors, in line with prior work, we propose this might be interpreted as identity encoding with enhanced contrast." eliminating "rather than valence encoding" since that is a speculation best reserved for discussion as the authors nicely do.

The above items stated, one issue comes to mind, and that is, why of all reasons would the authors find that the anteromedial aspect of the tubercle is not greatly reflecting valence. the anteromedial aspect of the tubercle, over all other aspects of the tubercle, is thought my many to more greatly partake in valence and other hedonic-driven behaviors given its dense reception of VTA DAergic fibers (as shown by Ikemoto, Kelsch, Zhang, and others). So this finding is paradoxical in contrast to if the authors would had studied the anterolateral tubercle or posterior lateral tubercle which gets less DA input.

---

## [Referee Report · Reviewer #3 (Public Review)]

Summary:

This manuscript describes a study of the olfactory tubercle in the context of reward representation in the brain. The authors do so by studying the responses of OT neurons to odors with various reward contingencies and compare systematically to the ventral pallidum. Through careful tracing, they present convincing anatomical evidence that the projection from the olfactory tubercle is restricted to the lateral portion of the ventral pallidum.

Using a clever behavioral paradigm, the authors then investigate how D1 receptor- vs. D2 receptor-expressing neurons of the OT respond to odors as mice learn different contingencies. The authors find that, while the D1-expressing OT neurons are modulated marginally more by the rewarded odor than the D2-expressing OT neurons as mice learn the contingencies, this modulation is significantly less than is observed for the ventral pallidum. In addition, neither of the OT neuron classes shows conspicuous amount of modulation by the reward itself. In contrast, the OT neurons contained information that could distinguish odor identities. These observations have led the authors to conclude that the primary feature represented in the OT may not be reward.

Strengths:

The highly localized projection pattern from olfactory tubercle to ventral pallidum is a valuable finding and suggests that studying this connection may give unique insights into the transformation of odor by reward association.

Comparison of olfactory tubervle vs. ventral pallidum is a good strategy to further clarify the olfactory tubercle's position in value representation in the brain.

Weaknesses:

The study comes to a different conclusion about the olfactory tubercle regarding reward representations from several other prior works. Whether this stems from a difference in the experimental configurations such as behavioral paradigms used or indeed points to a conceptually different role for the olfactory tubercle remains to be seen.

---

## [Author Response]

The following is the authors’ response to the previous reviews.

**Public Reviews:**

**Reviewer #1 (Public Review):**
In this manuscript, Lee et al. compared encoding of odor identity and value by calcium signaling from neurons in the ventral pallidum (VP) in comparison to D1 and D2 neurons in the olfactory tubercle (OT).Strengths:They utilize a strong comparative approach, which allows the comparison of signals in two directly connected regions. First, they demonstrate that both D1 and D2 OT neurons project strongly to the VP, but not the VTA or other examined regions, in contrast to accumbal D1 neurons which project strongly to the VTA as well as the VP. They examine single unit calcium activity in a robust olfactory cue conditioning paradigm that allows them to differentiate encoding of olfactory identity versus value, by incorporating two different sucrose, neutral and air puff cues with different chemical characteristics. They then use multiple analytical approaches to demonstrate strong, low-dimensional encoding of cue value in the VP, and more robust, high-dimensional encoding of odor identity by both D1 and D2 OT neurons, though D1 OT neurons are still somewhat modulated by reward contingency/value. Finally, they utilize a modified conditioning paradigm that dissociates reward probability and lick vigor to demonstrate that VP encoding of cue value is not dependent on encoding of lick vigor during sucrose cues, and that separable populations of VP neuros encode cue value/sucrose probability and lick vigor. Direct comparisons of single unit responses between the two regions now utilize linear mixed effects models with random effects for subject,Weaknesses:The manuscript still includes mention of differences in effect size or differing "levels" of significance between VP and OT D1 neurons without reports of a direct comparisons between the two populations. This is somewhat mitigated by the comprehensive statistical reporting in the supplemental information, but interpretation of some of these results is clouded by the inclusion of OT D2 neurons in these analyses, and the limited description or contextualization in the main text.

We think the reviewer is mistaken and have clarified the text. Each pairwise comparison between VP, OTD1 and OTD2, for each odor across days is shown as a heatmap in supplementary figure 8B, with further details in table 37. Absolute diff 3H no statistics

**Reviewer #2 (Public Review):**
We appreciate the authors revision of this manuscript and toning down some of the statements regarding "contradictory" results. We still have some concerns about the major claims of this paper which lead us to suggest this paper undergo more revision as follows since, in its present form, we fear this paper is misleading for the field in two areas. here is a brief outline:(1) Despite acknowledging that the injections only occurred in the anteromedial aspect of the tubercle, the authors still assert broad conclusions regarding where the tubercle projects and what the tubercle does. for instance, even the abstract states "both D1 and D2 neurons of the OT project primarily to the VP and minimally elsewhere" without mention that this is the "anteromedial OT". Every conclusion needs to specify this is stemming from evidence in just the anteromedial tubercle, as the authors do in some parts of the the discussion.

We have clarified in multiple locations that we are recorded from the anteromedial OT, including the abstract, and further clarified this in the conclusions throughout the results and discussion. We refrain stating “anteromedial OT” at every mention of the OT, but think we have now made it abundantly clear that our observations are from the anteromedial OT. It is worth noting that retrograde tracing from the VTA did not label any neuron in any part of the OT, suggesting that the conclusion may well extend beyond the anteromedial portion. Though, we acknowledge further work is needed to comprehensively characterize the OT outputs.

(2) The authors now frame the 2P imaging data that D1 neuron activity reflects "increased contrast of identity or an intermediate and multiplexed encoding of valence and identity". I struggle to understand what the authors are actually concluding here. Later in discussion, the authors state that they saw that OT D1 and D2 neurons "encode odor valence" (line 510).

The point we aim to make is that valence encoding is different between the OT and VP. We do not think the reward modulated activity in OT is valence encoding, at least not as it is in the VP. We do observe some valence encoding at the population level, which is different from individual valence encoding neurons. The ability of classifiers to segregate population activity based on reward might be considered valence encoding, but we contrast it with that in VP where individual neurons signal reward prediction. This is more robust than that in the OT data where few neurons robustly encode valence. The increased response of the OTD1 neurons after reward association, is more consistent with contrast enhancement than valence encoding. We believe this distinction is important and reflects a transformation between two reward-related brain areas. For clarification of the sentence in question we have changed it to reflects “increased contrast of iden-ty or an intermediate encoding of valence that also encodes iden-ty.” (line 488)

We appreciate the authors note that there is "poor standardization" when it comes to defining valence (line 521). We are ok with the authors speculating and think this revision is more forthcoming regarding the results and better caveats the conclusions. I suggest in abstract the authors adjust line 14/15 to conclude that, "While D1 OT neurons showed larger responses to rewarded odors, in line with prior work, we propose this might be interpreted as identity encoding with enhanced contrast." eliminating "rather than valence encoding" since that is a speculation best reserved for discussion as the authors nicely do.

We accept this suggestion and have modified the abstract sentence to say, “Though D1 OT neurons showed larger responses to rewarded odors than other odors, consistent with prior findings, we interpret this as iden-ty encoding with enhanced contrast.” We believe this is appropriately qualified as an interpreta-on, and should not be confusing.

The above items stated, one issue comes to mind, and that is, why of all reasons would the authors find that the anteromedial aspect of the tubercle is not greatly reflecting valence. the anteromedial aspect of the tubercle, over all other aspects of the tubercle, is thought my many to more greatly partake in valence and other hedonic-driven behaviors given its dense reception of VTA DAergic fibers (as shown by Ikemoto, Kelsch, Zhang, and others). So this finding is paradoxical in contrast to if the authors would had studied the anterolateral tubercle or posterior lateral tubercle which gets less DA input.

We agree that this seems surprising. This is why we focused on the anteromedial expecting to find valence encoding. It remains possible that other parts of the OT, or more dorsal aspects of the anteromedial OT encode valence, as has been reported by Murthy and colleagues. However, it remains unclear if their recordings are in the OT or VP. Nonetheless our findings indicate that more work is required to understand the contribution of the OT to valence encoding. It is also important to note that our conclusions are drawn in comparison to the VP, which has more robust valence encoding than the OT. Thus, in comparison the OT sample in our recordings lack robust valence signaling. We think this comparison is important, due to the lack of clear framework for defining valence that may create misleading statements in past OT work.

**Reviewer #3 (Public Review):**
Summary:This manuscript describes a study of the olfactory tubercle in the context of reward representation in the brain. The authors do so by studying the responses of OT neurons to odors with various reward contingencies and compare systematically to the ventral pallidum. Through careful tracing, they present convincing anatomical evidence that the projection from the olfactory tubercle is restricted to the lateral portion of the ventral pallidum.Using a clever behavioral paradigm, the authors then investigate how D1 receptor- vs. D2 receptor-expressing neurons of the OT respond to odors as mice learn different contingencies. The authors find that, while the D1-expressing OT neurons are modulated marginally more by the rewarded odor than the D2-expressing OT neurons as mice learn the contingencies, this modulation is significantly less than is observed for the ventral pallidum. In addition, neither of the OT neuron classes shows conspicuous amount of modulation by the reward itself. In contrast, the OT neurons contained information that could distinguish odor identities. These observations have led the authors to conclude that the primary feature represented in the OT may not be reward.Strengths:The highly localized projection pattern from olfactory tubercle to ventral pallidum is a valuable finding and suggests that studying this connection may give unique insights into the transformation of odor by reward association.Comparison of olfactory tubervle vs. ventral pallidum is a good strategy to further clarify the olfactory tubercle's position in value representation in the brain.Weaknesses:The study comes to a different conclusion about the olfactory tubercle regarding reward representations from several other prior works. Whether this stems from a difference in the experimental configurations such as behavioral paradigms used or indeed points to a conceptually different role for the olfactory tubercle remains to be seen.

We acknowledge that our results lead us to conclusions that are different from that of prior work. But we note that our results are not directly at odds, as we see similar reward modulation of D1 OT neurons as has been reported previously. Our conclusion is different because we contrast our OT responses with that in the VP where valence is more robustly encoded at the single neuron level. We also note, that many of the past studies do not define valence as stringently as we do. Thus, increased activity with reward, as observed in our data and past studies, seems more like reward modulation than valence.